# PACT prevents aberrant activation of PKR by endogenous dsRNA without sequestration

Sadeem Ahmad[1,2,10], Tao Zou[3,4,10], Jihee Hwang[2,5,10], Linlin Zhao [2,8,10], Xi Wang[1,2], Anton Davydenko[1,2], Ilana Buchumenski[6], Patrick Zhuang[3,4], Alyssa R. Fishbein[3,4], Diego Capcha-Rodriguez [3,4], Aaron Orgel[3,4], Erez Y. Levanon [6], Sua Myong [2,5], James Chou [2,9] ✉, Matthew Meyerson [3,4,7] ✉ & Sun Hur [1,2] ✉

The innate immune sensor PKR for double-stranded RNA (dsRNA) is critical for antiviral defense, but its aberrant activation by cellular dsRNA is linked to various diseases. The dsRNA-binding protein PACT plays a critical yet controversial role in this pathway. We show that PACT directly suppresses PKR activation by endogenous dsRNA ligands, such as inverted-repeat Alu RNAs, which robustly activate PKR in the absence of PACT. Instead of competing for dsRNA binding, PACT prevents PKR from scanning along dsRNA—a necessary step for PKR molecules to encounter and phosphorylate each other for activation. While PKR favors longer dsRNA for increased co-occupancy and scanning-mediated activation, longer dsRNA is also more susceptible to PACT-mediated regulation due to increased PACT-PKR co-occupancy. Unlike viral inhibitors that constitutively suppress PKR, this RNA-dependent mechanism allows PACT to fine-tune PKR activation based on dsRNA length and quantity, ensuring self-tolerance without sequestering most cellular dsRNA.

Effective antiviral defense relies on several innate immune receptors that detect viral dsRNAs and trigger antiviral immune responses[1,2]. These receptors include a protein kinase R (PKR) that suppresses global protein synthesis upon recognizing dsRNA, RIG-I-like receptors (RLRs) that activate the type I interferon signaling pathway, and oligoadenylate synthases (OASes) responsible for activating RNase L, leading to the widespread degradation of both cellular and viral RNAs[1,2]. While these dsRNA receptors work in synergy to mount effective antiviral immune responses, recent studies have brought to light the unique role of PKR as a key determinant of cellular fate in response to dsRNA, in particular endogenous dsRNA accumulating

under dysregulated conditions[3–10]. Studies have also shown that PKR is a major contributor to the pathogenesis of a broad range of neurodevelopmental and auto-inflammatory diseases, including Parkinson's disease[11–14], systemic lupus erythematosus (SLE)[15] and proteasome-associated autoinflammatory syndromes (PRAAS)[16], although the underlying molecular mechanisms for the aberrant PKR activation under these disease conditions and its tissue specificity are only beginning to be understood[17,18].

PKR represents one of the four kinases that initiate integrated stress response (ISRs), a conserved cellular response to common stressors, including oxidative stress, nutrient starvation, and viral

[1]Howard Hughes Medical Institute and Program in Cellular and Molecular Medicine, Boston Children's Hospital, Boston, MA, USA. [2]Department of Biological Chemistry and Molecular Pharmacology, Harvard Medical School, Boston, MA, USA. [3]Department of Medical Oncology, Dana-Farber Cancer Institute, Boston, MA, USA. [4]Cancer Program, Broad Institute of Harvard and MIT, Cambridge, MA, USA. [5]Program in Cellular and Molecular Medicine, Boston Children's Hospital, Harvard Medical School, Boston, MA, USA. [6]Mina and Everard Goodman Faculty of Life Sciences, Bar-Ilan University, Ramat Gan, Israel. [7]Department of Genetics, Harvard Medical School, Boston, MA, USA. [8]Present address: Tongji University Cancer Center, Shanghai Tenth People's Hospital, School of Medicine, Tongji University, Shanghai, China. [9]Present address: Interdisciplinary Research Center on Biology and Chemistry, Shanghai Institute of Organic Chemistry, Chinese Academy of Sciences, Shanghai, China. [10]These authors contributed equally: Sadeem Ahmad, Tao Zou, Jihee Hwang, Linlin Zhao. ✉e-mail: james_chou@mail.sioc.ac.cn; matthew_meyerson@dfci.harvard.edu; sun.hur@crystal.harvard.edu

infection[19]. In the resting state, PKR is in a latent, unphosphorylated, and monomeric state. Upon dsRNA binding, however, PKR undergoes dimerization and autophosphorylation, both of which cooperate to activate its kinase activity[20,21]. In this process, dsRNA serves as a scaffold to bring together PKR molecules, facilitating their dimerization and inter-molecular phosphorylation[22,23]. This notion is supported by observations that PKR activation requires a minimum of ~30 bp-long dsRNA, sufficient to accommodate two PKR molecules, and that PKR is inhibited by an excess amount of dsRNA, which disperses PKR among many dsRNA molecules, preventing autophosphorylation between PKR molecules[23,24]. Once activated, PKR phosphorylates the translational initiation factor eIF2α, which then results in the inhibition of translational initiation for the majority of capped mRNAs except for a small group of stress-related genes that help the cell recover from stress conditions[25,26]. While transient activation of ISR is beneficial for the cell, its prolonged activation can lead to cellular toxicity through various mechanisms[19,27].

The importance of PKR in cell fate decision is underscored by the multi-layered regulatory mechanisms for PKR. One extensively studied negative regulator is ADAR1, an RNA editing enzyme that converts adenosines (A) to inosines (I) within duplex RNA. A-to-I modification typically weakens Watson-Crick base-pairing, disrupting dsRNA structure[28]. This disruption plays a crucial role in preventing inappropriate activation of multiple dsRNA sensors in the cell, including PKR, and averting unwanted innate immune system activation[5-10]. More recent studies suggested that ADAR1 can also suppress PKR in a manner independent of the catalytic activity[6], implicating the importance of its RNA binding activity, in addition to editing functions.

PACT is another protein suggested to modulate PKR and other dsRNA sensing pathways, but its functions and mechanisms have been controversial. Widely considered a PKR activator[29-32] --thus named PACT (encoded by *PRKRA*)-- more recent genetic studies contradicted this prevailing notion. Knocking out the PACT homolog (RAX) in mice causes reproductive and developmental defects, which were rescued by PKR double knock-out (DKO)[33]. Impaired expression of PACT in 293 T or HeLa cells[34,35] also showed increased activity of PKR. Although the genetic evidence suggested the PKR-suppressive functions of PACT[33-35], the underlying mechanisms--including whether such effects are direct or indirect--have been unclear, largely because these inhibitory functions have not been biochemically reconstituted. In fact, earlier biochemical studies suggested that PACT may activate PKR independent of dsRNA[29,30,36], raising questions about the exact functions of PACT and how these findings can be reconciled altogether.

In contrast to PACT's seemingly complex and controversial biological functions, its domain architecture is remarkably simple, featuring just three tandem repeats of dsRNA-binding domains (DRBDs). DRBDs are common in dsRNA-binding proteins across all life kingdoms and viruses, with a conserved fold and mode of dsRNA interaction[37,38]. DRBDs commonly recognize dsRNA backbone structure with little dependence on dsRNA sequence[38]. However, certain DRBDs have evolved to function as protein-protein interaction domains at the expense of dsRNA-binding activity[39-42]. For PACT, the first two DRBDs exhibit typical sequence-independent dsRNA-binding activity. The third DRBD, however, shows little to no dsRNA binding activity and instead forms a homodimer[43,44], showcasing a divergent evolution of DRBD as a protein-interaction domain.

We here examine how PACT modifies the PKR activity using a combination of biochemistry, single-molecule analyzes, cellular functional studies, and structure-guided mutagenesis. Our data unambiguously indicate that PACT is a direct and specific negative regulator of PKR, restricting PKR's movement on dsRNA and limiting its ability to undergo autophosphorylation without blocking its dsRNA binding per se. This reveals a unique mode of regulating foreign nucleic acid sensors in innate immunity.

## Results

### PACT restricts aberrant activation of PKR, thereby maintaining cellular homeostasis

Previous studies showed that aberrant activation of PKR leads to the loss of cell fitness in many cancer cell lines. To investigate the role of PACT in PKR functions, we examined the impact of PACT depletion using 10 different cancer cell lines. In 5 out of 10 cells tested (HCC1806 - breast, MDA-MB-157 - breast, NCI-H727 - lung, NCI-H2286 - lung, SW-1271 - lung), knocking out PACT led to a loss of cell fitness, as measured by ATP bioluminescence assay (Fig. 1A). We chose three of these cell lines—HCC1806, NCI-H727, and NCI-H2286—and confirmed their sensitivity to PACT depletion using crystal violet staining, an orthogonal cell viability assay (Fig. 1B). This loss of cell fitness could be prevented by overexpressing wild-type PACT in PACT-deficient HCC1806 cells (Fig. 1C), further supporting the role that PACT as an essential gene.

To examine whether PKR is aberrantly activated by PACT depletion, we measured the levels of phospho-PKR (p-PKR) and phospho-eIF2α (p-eIF2α) by western blot (WB). The results showed that PACT dependency segregated with PKR activation. PACT depletion increased PKR activity in PACT knockout (KO)-sensitive cells, but not in PACT KO-insensitive cells (Fig. 1D). Similarly, PACT depletion led to the induction of *CHOP*, a known consequence of eIF2α phosphorylation[45], in PACT KO-sensitive cells, but not in PACT KO-insensitive cells (Supplementary Fig. 1A). Consistent with the notion that aberrant activation of PKR caused increased eIF2α phosphorylation and loss of cell fitness, double knockout (DKO) of PKR returned p-eIF2α to the basal level and rescued cell viability (Figs. 1E, F and Supplementary Fig. 1B–D). Interestingly, ISRIB, a small molecule inhibitor of ISR[46], showed varying degrees of partial rescue in 2 different PACT-dependent cell lines, suggesting that PKR regulates cell fitness through both ISR-dependent and -independent mechanisms (Fig. 1G and Supplementary Fig. 1E, F).

We then examined how other dsRNA-dependent innate immune pathways are affected by PACT depletion and whether PACT's inhibitory function is specific to PKR. To test OAS activity, we measured the integrity of rRNAs, which are well-known targets of RNase L downstream of OASes. Bioanalyzer data showed that rRNAs are intact in all cells, suggesting minimal activation of OASes in the absence of PACT regardless of cell lines (Supplementary Fig. 1G). Furthermore, we did not observe RLR activation, as measured by *IFNβ* and *IL-6* induction, in most cells, except for the HCC1806 cell line (Supplementary Fig. 1A). Consistent with these observations, knocking out RNase L or the RLR adaptor MAVS had little to minor impact on either cell viability or PKR activity (Supplementary Fig. 2A–D). Altogether, these results suggest that PACT depletion leads to aberrant activation of PKR and consequent loss of cell fitness, without necessarily activating other dsRNA-dependent innate immune signaling pathways.

### PACT shares dsRNA ligands with PKR, but does not sequester dsRNA away from PKR

We next examined how PACT inhibits PKR. Since PACT is a dsRNA-binding protein ($K_d$ of 357.5 nM), we asked whether PACT's dsRNA-binding activity is important. We generated the dsRNA binding-deficient variant that harbors four Lys-to-Glu mutations (4KE) in the dsRNA interface of DRBD1 (K84, K85) and DRBD2 (K177, K178) (Fig. 2A). Complementation with WT PACT in PACT-depleted HCC1806 cells restored cell viability, but 4KE PACT did not (Fig. 2B), indicating that PACT requires dsRNA binding to regulate PKR.

One possible mechanism is that PACT sequesters endogenous dsRNA, thereby limiting their access to PKR. To test this possibility, we examined whether PACT and PKR share their dsRNA ligands in cells and whether the presence of PACT affects PKR–dsRNA interaction. To measure the interactions between PKR/PACT and dsRNA in cells, we employed the formaldehyde-assisted RNA-protein crosslinking and immunoprecipitation (fCLIP), a method well-established for capturing

protein–dsRNA interactions that are often difficult to capture using direct UV crosslinking methods[47].

We performed PKR fCLIP in PACT-depleted HCC1806 cells, one of the PACT KO-sensitive cell lines. Because PACT deletion results in significant compromise in cell viability, we generated a PACT/PKR DKO model and complemented it with the HA-tagged, kinase-dead PKR mutant (K296R, Supplementary Fig. 3A), which does not impair cell viability. Our initial validation of PKR fCLIP demonstrated significant enrichment of an in vitro transcribed 112 bp dsRNA transfected into the cells, exhibiting ~500-fold increase over the input control (Supplementary Fig. 3B). Anti-HA showed a cleaner background signal than

anti-PKR in 112 bp dsRNA enrichment (Supplementary Fig. 3B) and therefore was chosen for the subsequent PKR fCLIP experiments. We next proceeded with PKR fCLIP without the exogenously introduced dsRNA to examine the endogenous dsRNA ligands. As a control, we also performed fCLIP in the absence of HA-PKR. Among 116 PKR fCLIP peaks (Log2FC ≥ 1) (Supplementary Data 1), we computationally removed 12 peaks, predominantly rRNAs, which were also enriched by fCLIP in the absence of HA-PKR (Log2FC > 0) (Supplementary Fig. 3C). This filtering led to 94 PKR-specific fCLIP peaks (Fig. 2C and Supplementary Data 2), of which 79 peaks were invert-repeat Alu RNAs (IR-Alus) and regions proximal to IR-Alu pairs (within 300 bp of or

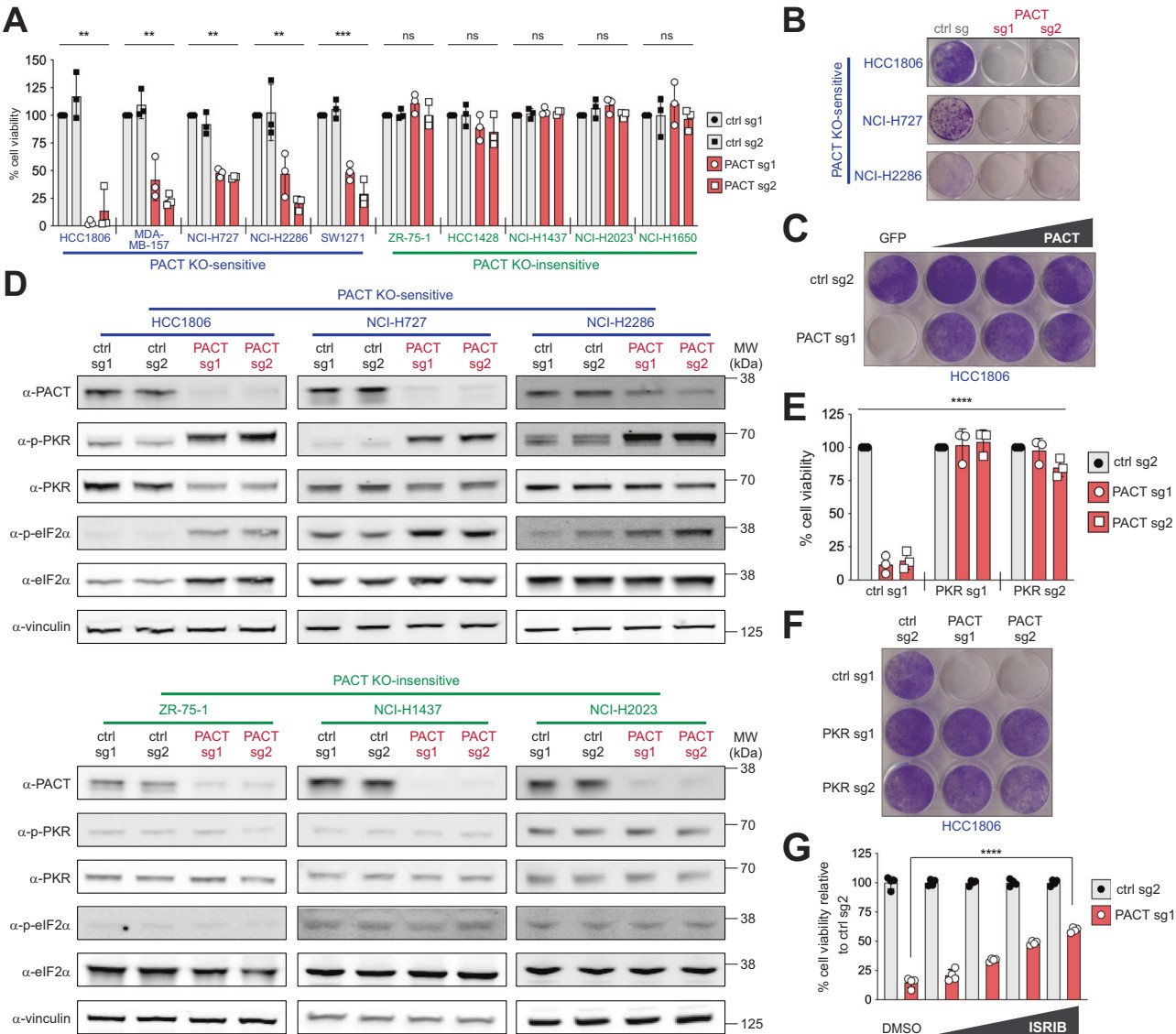

**Fig. 1 | PACT restricts aberrant activation of PKR, thereby maintaining cellular homeostasis. A** ATP bioluminescence assay showing cell viability in PACT KO-sensitive (HCC1806, MDA-MB-157, NCI-H727, NCI-H2286, SW1271) and -insensitive (ZR-75-1, HCC1428, NCI-H1437, NCI-H2023, NCI-H1650) cell lines after control gene-KO (ctrl) or PACT-KO using 2 different guide RNAs. The control (ctrl) genes are AAVS1 (sg1) and Chr2.2 (sg2). Values represent means ( ± SD) of 3 biological repeats. *P* values were based on one-way ANOVA test. **B** Crystal violet staining assay showing cell viability in 3 PACT KO-sensitive cells (HCC1806, NCI-H727, and NCI-H2286) after control gene-KO vs. PACT-KO. **C** Crystal violet staining assay with HCC1806 cells after control gene-KO and PACT-KO complemented with increasing amounts of wild-type PACT. PACT used here was engineered to be resistant to PACT sgRNA 1 targeting. **D** Western blot analysis showing levels of PKR and eIF2α phosphorylation

in PACT KO-sensitive (top panel) and -insensitive (bottom panel) cells after control gene-KO or PACT-KO. Vinculin was used as a loading control. The experiment was repeated 3 times independently with similar results. **E** ATP bioluminescence assay in HCC1806 cells in PKR-sufficient and PKR-KO cells in control gene-KO and PACT-KO backgrounds. Values represent means ( ± SD) of 3 biological repeats. *P* values were based on two-way ANOVA test. **F** A representative crystal violet staining for samples in **E**. **G** ATP bioluminescence assay in HCC1806 cells in control gene-KO and PACT-KO treated with indicated concentrations of ISRIB. Values represent means ( ± SD) from 4 technical replicates across two independent experiments (*n* = 2). *P* values were based on two-tailed student's *t*-test. ****p = <0.0001, ***p = <0.001, **p = <0.01, *p = <0.05, not significant (ns) is for *p* > 0.05. Source data and exact *P* values are provided as a Source Data file.

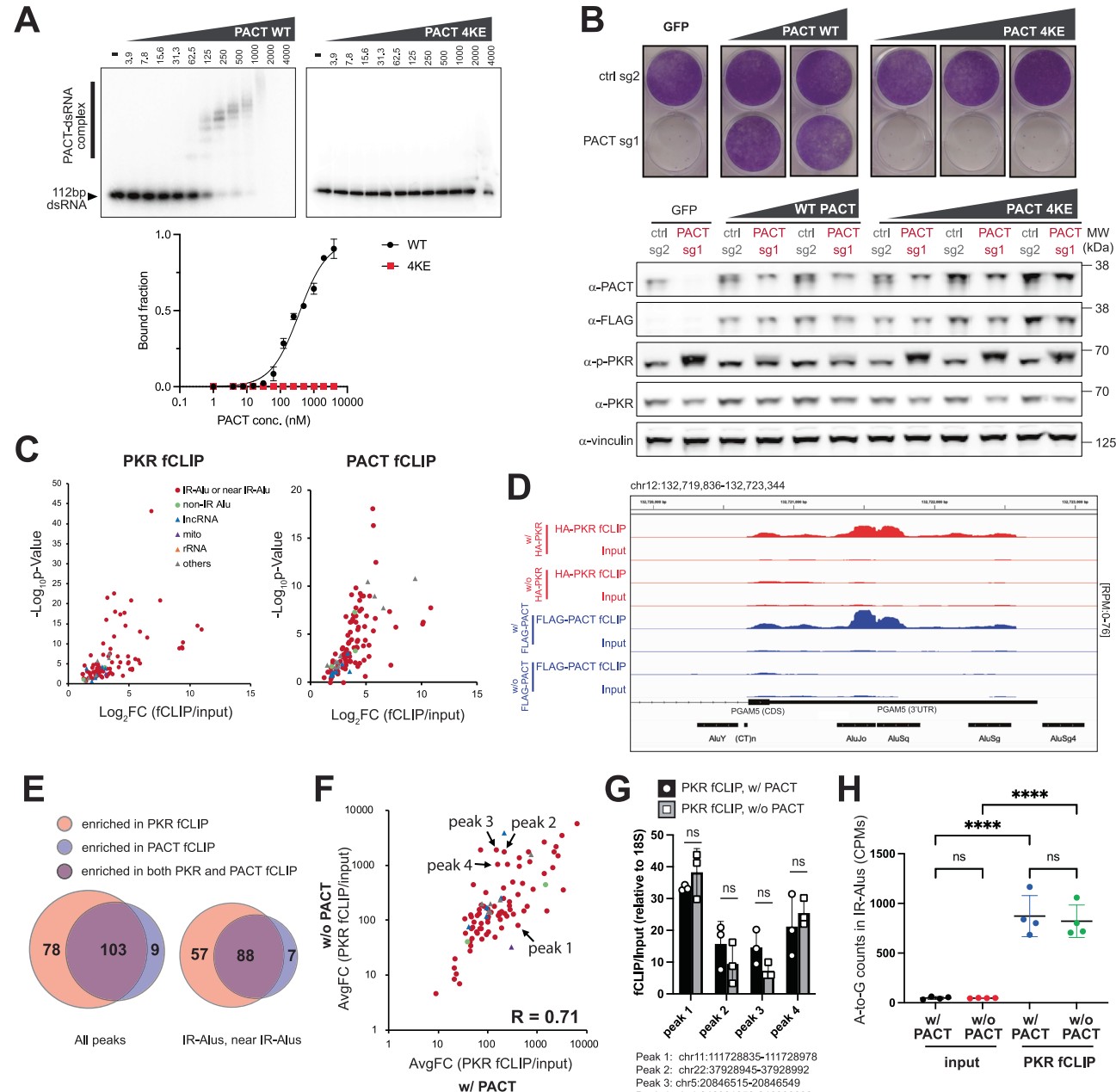

**Fig. 2 | PACT and PKR share endogenous dsRNA ligands, but PACT inhibits PKR without blocking PKR's dsRNA binding. A** Gel-shift assay with 112 bp dsRNA binding to increasing concentrations (0–4000 nM) of PACT wild-type or 4KE (K84E + K85E + K177E + K178E) mutant. Right: specific binding curve using Hill equation assuming 8 binding sites (see "Methods"). Wild-type plot represents mean ± SD from 3 experiments. **B** Crystal violet assay showing cell viability (HCC1806) after control/PACT-KO, complemented with increasing expression of PACT wild-type or 4KE. Bottom: WB analysis. The experiment was repeated 3 times. **C** Volcano plots showing RNA peaks enriched by PKR (left) and PACT (right) fCLIP in HCC1806 cells. Peak identification criteria: (Log$_2$FC (HA-fCLIP/input) ≥1 with HA-PKR & ≤0 without HA-PKR) and (Log$_2$FC (FLAG-fCLIP/input) ≥1 with FLAG-PACT & ≤0 without FLAG-PACT). See also Supplementary Fig. 3C, F. Mean from 3 (PKR) and 2 (PACT) replicates. *P* values were calculated in Deseq2 (two-sided) and adjusted by Benjamini-Hochberg method. **D** Snapshot of an IR-Alu sequence enriched in PKR (red) and PACT (blue) fCLIP-seq. **E** Venn diagrams showing overlap of total (left)

and IR-Alu/near IR-Alu peaks (right) enriched in PKR and PACT fCLIP. Near IR-Alu denote peaks flanked by or adjacent to IR-Alus (<300 bp). **F** HA-PKR fCLIP intensity before (*x*-axis) and after PACT KO (*y*-axis). Plot shows the average fold change of area under the curve (AuC) for peaks from (**C**). RNA levels (AuC) were normalized to total non-rRNA reads, and fCLIP/input ratio calculated for normalized AuC. See Supplementary Fig. 4A and Methods for additional details. Mean from 2 biological repeats. Correlation coefficient (*R*) = 0.71. **G** fCLIP-RT-qPCR validation of 4 peaks from (**F**) using endogenous PKR. RNA levels were normalized to 18S rRNA before calculating fCLIP/input ratio. Mean (± SD) from 3 replicates. *P* values from a two-tailed *t*-test. (ns, not significant; *P* > 0.05). **H** A-to-I editing levels in PKR fCLIP and input samples (HCC1806) before and after PACT KO, measured as A-to-G mismatch counts per million reads (CPM). Mean (± SD) of 4 biological repeats. *P* values from one-way ANOVA followed by Tukey multiple comparisons test. ****$p$ < 0.0001; (ns) $p$ > 0.12. Source data and exact *P* values are provided as a Source Data file.

flanked by IR-Alu pair) (see Fig. 2C, D). This is consistent with the previous reports showing that IR-Alus are the primary endogenous dsRNAs in primates, and serve as the major ligands for PKR and other dsRNA binding proteins[4,5,48,49]. Unlike a previous report showing

association between PKR and sense-antisense overlapping transcripts from mitochondria in HeLa cells[3], we observed little sign of sense-antisense overlaps in mitochondrial transcripts either in our input or PKR fCLIP samples (Supplementary Fig. 3D). This discrepancy may

reflect cell-type-dependent variability in mitochondrial dsRNA accumulation, as previously noted[17].

To identify RNA ligands for PACT, we performed a similar PACT fCLIP using HCC1806 cells ectopically expressing FLAG-tagged PACT. Anti-FLAG fCLIP also showed robust enrichment of exogenously introduced 112 bp dsRNA (Supplementary Fig. 3E), validating our approach. PACT fCLIP showed RNA ligands that largely overlapped with those identified from PKR fCLIP (Fig. 2C–E and Supplementary Fig. 3F, Supplementary Data 3, Supplementary Data 4). Specifically, IR-Alu pairs and regions proximal to IR-Alus, identified from PKR fCLIP, represented the majority of the PACT-bound RNA species (Fig. 2E, see also Fig. 2D). As with PKR fCLIP, we observed little enrichment of mitochondrial RNAs (Supplementary Fig. 3D).

Given that PACT and PKR share the endogenous dsRNA ligands, we investigated whether PACT competes with PKR for the shared dsRNA substrates. To address this, we compared PKR fCLIP intensity from PACT-sufficient and -deficient cells, and found that most PKR peaks were largely unaffected by PACT (Fig. 2F and Supplementary Fig. 4A for normalization validation, Supplementary Data 5). While a small subset of PKR fCLIP peaks showed variable changes in intensity upon PACT depletion, their changes were statistically insignificant as observed by fCLIP RT-qPCR (Supplementary Fig. 4B). Furthermore, comparison of fCLIP RT-qPCR with endogenous PKR corroborated this finding (Fig. 2G).

We also examined whether ADAR1-mediated A-to-I editing of dsRNA, known to affect PKR activity, was somehow affected by PACT. However, the A-to-I editing level of either PKR fCLIPed IR-Alus or overall IR-Alus were not affected by the presence of PACT (Fig. 2H and Supplementary Fig. 4C).

These results suggest that PACT regulates PKR without limiting PKR's access to endogenous dsRNA, and this regulation is independent of ADAR1. Additionally, given that PACT knockout primarily affects PKR but not ADAR1, OASes-RNase L, and RLRs (in two out of three cell lines tested for RLRs) (Supplementary Fig. 1A), we conclude that PACT can inhibit PKR without globally sequestering cellular dsRNAs.

## PKR scans along dsRNA, resulting in dsRNA length-dependent activity

To investigate the molecular mechanism by which PACT inhibits PKR, we performed an in vitro PKR activity assay, measuring PKR autophosphorylation using radiographic [$\gamma$-$^{32}$P]-ATP incorporation. PKR activity was quantified on SDS-PAGE, with [$^{35}$S]-labeled IRF3 serving as a loading control (Fig. 3A).

First, we assessed PKR activity in the absence of PACT with various dsRNA sequences, lengths, and concentrations. Consistent with previous studies[50], PKR was not significantly influenced by dsRNA sequence or GC content (Fig. 3A). However, PKR activity strongly depended on dsRNA length. While PKR is known to require dsRNA lengths >30 bp for activation[23], 42 bp dsRNA showed only weak activity, despite the monomeric PKR footprint of 15 bp (Supplementary Figs. 5A, B). In comparison, 62 bp dsRNA led to robust activation, and 112 bp dsRNA induced an even stronger response (Fig. 3B–D). This length dependence persisted regardless of whether activity was compared by RNA mass (Fig. 3D) or molar concentration (Supplementary Fig. 5C). Importantly, increasing the dose of shorter dsRNAs, such as 62 bp, could not elevate PKR activity to levels observed with 112 bp dsRNA (Fig. 3D).

Next, we investigated how PKR responds to its endogenous ligands, IR-Alus, and the effect of structural irregularities, such as mismatches and bulges, within IR-Alus. PKR activity with 112 bp dsRNA was similar to that of 388 bp perfect Alu dsRNA (Fig. 3E), suggesting that PKR's length sensitivity saturates around 100–400 bp. IR-Alu derived from the 3′UTRs of NICN1 and PSMB2 genes, which harbor ~20% mismatches and bulges (Supplementary Fig. 5D), were less efficient at activating PKR than perfect dsRNA of the same length (Fig. 3E),

suggesting that PKR activity is reduced by mismatches and bulges. Nevertheless, IR-Alus still triggered over 60% of the activation seen with 388 bp perfect dsRNA across all RNA concentrations tested, supporting their potential to stimulate PKR.

We then explored whether PKR's length-dependent activation could be due to multimerization along dsRNA, as seen with other dsRNA sensors like MDA5[51,52]. However, negative-stain EM revealed no stable PKR multimerization along dsRNA (Supplementary Fig. 5E). Alternatively, we hypothesized that PKR might undergo one-dimensional diffusion along dsRNA, as seen with other proteins containing tandem repeats of dsRNA-binding domains, like PACT, TRBP, and Staufen 1[53–55]. Given that PKR cannot autophosphorylate intramolecularly[26], this scanning activity may promote PKR's interaction with each other and inter-molecular autophosphorylation, explaining dsRNA length-dependent activation.

To test this model, we conducted single-molecule fluorescence resonance energy transfer (FRET) measurements between Cy3-labeled 112 bp dsRNA (donor) and Cy5-labeled PKR (acceptor) (Fig. 3F, left). The dsRNA was immobilized on a surface via a biotin label, and PKR was introduced. We used catalytically inactive K296R PKR to analyze PKR movement before its activation. At a concentration of 50 nM, PKR bound to dsRNA infrequently, typically exhibiting one or two binding events (FRET > ~0.6 and anticorrelated Cy5 and Cy3 signal changes) over ~30 seconds (s). While some binding events were isolated, lasting <2 s, (e.g., the event marked with * in Fig. 3F), many occurred in clusters ("On"), during which rapid FRET fluctuations were observed, similar to those previously reported for TRBP and PACT[53,54] (Fig. 3F and Supplementary Fig. 6B). Given the low frequency of binding events, these fluctuations are unlikely due to rapid association and dissociation. They also do not represent random noise, but likely reflect PKR scanning along dsRNA. Several observations support this interpretation: (1) During the rapid FRET fluctuations in "On" state, the fluorescence signals of the acceptor and donor were anticorrelated (Fig. 3F and Supplementary Fig. 6B), consistent with dynamic changes in the distance between PKR and dsRNA end. (2) FRET fluctuations in the "On" state also displayed autocorrelation with a single exponential decay (Fig. 3G), a behavior not seen in the "Off" state (Fig. 3G) or without PKR (Supplementary Fig. 6C, D). (3) The sliding time of PKR (inverse of the decay rate from the autocorrelation analysis) increased with dsRNA length (Fig. 3H and Supplementary Fig. 6E), a characteristic expected for PKR movement along dsRNA, but not for random noise. (4) Similar fluctuations were observed by protein-induced fluorescence enhancement (PIFE) analysis upon PKR addition (Supplementary Fig. 7A). These observations collectively support the notion that PKR rapidly scans along dsRNA in a length-dependent manner.

Notably, similar scanning behaviors were observed with PACT in this study (Supplementary Fig. 7B–E) and in earlier reports[53,54]. These findings align with the known scanning activities of other dsRNA-binding proteins, such as TRBP and Staufen 1[53–55]. However, not all dsRNA-binding proteins exhibit this behavior. For instance, Dicer and ADAR1/2 show little evidence of scanning on dsRNA[54,55], providing useful reference points for comparison with PKR. Collectively, our findings suggest that PKR's diffusion along dsRNA—together with the requirement for intermolecular interactions for autophosphorylation,—explains the observed dsRNA length- and concentration-dependent activation.

## PACT inhibits PKR by restricting its motion on dsRNA, leading to length-dependent inhibition

We examined how PACT affects PKR activity by varying dsRNA lengths and concentrations and visualizing the results as heatmaps with increasing PACT levels (Fig. 4A, B, see Supplementary Fig. 8A for graphs). In all cases, PACT raised the dsRNA threshold for PKR activation, confirming its role as an inhibitor rather than an activator. This inhibitory effect was consistent across four different dsRNAs (varying

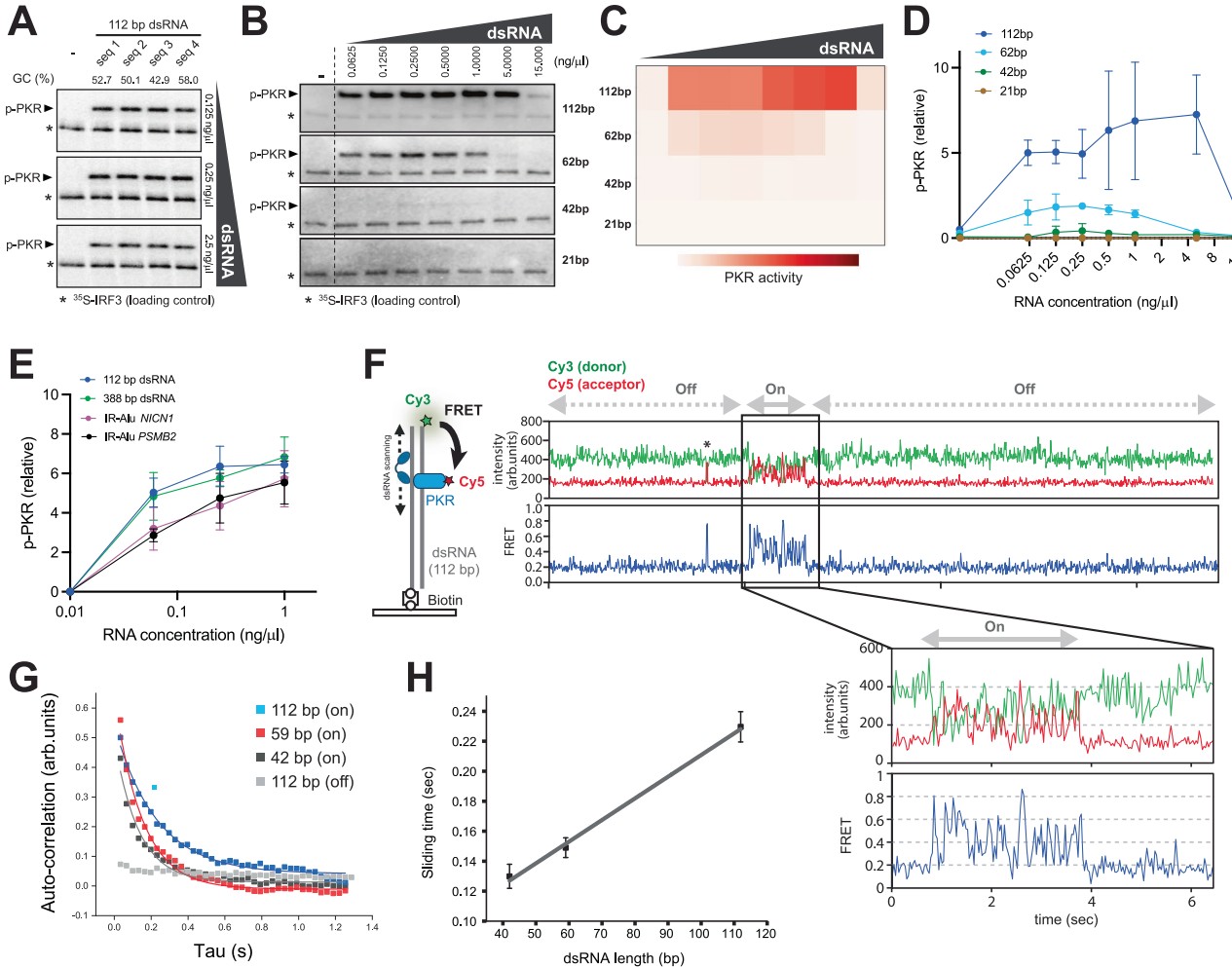

**Fig. 3 | PKR scans along dsRNA, resulting in dsRNA length-dependent activity.**
**A** In vitro PKR kinase assay monitoring PKR autophosphorylation with increasing concentrations (0.125, 0.25, 2.5 ng/μl) of 112 bp dsRNA with 4 different sequences (see supplementary Table 3) and GC contents. [$^{35}$S]-IRF3 served as loading control (*). Wild-type full-length PKR was used unless stated otherwise. Results were reproduced in 2 replicates. **B** PKR kinase assay with increasing dsRNA concentrations (0.0625–15 ng/μl) for lengths 112, 62, 42, and 21 bp. Results reproduced in 3 replicates. **C, D** Quantification of PKR activity from **A** displayed as heatmap and x-y graph. Data represent mean (± SD) of 3 biological repeats. **E** PKR activity with 112 bp dsRNA, perfect duplex Alu dsRNA, and IR-Alus from *NICN1* and *PSMB2* 3′-UTRs (see Supplementary Fig. 5D) at 0.0625, 0.25, and 1 ng/μl. Values are mean (± SD) of 3 biological replicates. **F** Schematic of single-molecule FRET experiments to monitor scanning motions of PKR on dsRNA. Cy5-labeled PKR (acceptor, 50 nM) was added to a chamber containing Cy3-labeled dsRNA (donor) immobilized on the surface.

Cy3 and Cy5 fluorescence were monitored upon Cy3 excitation using sm-TIRF microscopy. Time course traces of Cy3/Cy5 intensities and FRET (right panel) and zoomed-in trace for "On" state (lower panel). PKR K296R was used to ensure analysis of PKR movement before its activation. *indicates transient or abortive binding--such events were excluded from the autocorrelation analysis in **G**. See Supplementary Fig. 6 for additional traces and description of identifying "On" events. Created in BioRender. torres, c. (2025) https://BioRender.com/g96x368. **G** Autocorrelation analysis of FRET fluctuations for PKR in the "On" state on 112, 59, 42 bp dsRNAs and "Off" state on 112 bp dsRNA. See Methods for the autocorrelation function. (**H**) Diffusion rates (mean ± SD) calculated from exponential fitting of the autocorrelation curves of PKR (see Methods) with different dsRNA lengths: 0.130 ± 0.00657 s (42 bp), 0.149 ± 0.00657 s (59 bp) and 0.230 ± 0.0101 s (112 bp); n = 143 (42 bp), 69 (59 bp) and 236 (112 bp) diffusion events. Source data is provided as a Source Data file.

in sequence and GC-content) and IR-Alus (Supplementary Fig. 8B, C). We note that a robust and reproducible inhibitory effect of PACT was observed only after implementing an elaborate purification process to eliminate contaminating RNA from the bacterial expression system. We also confirmed that there is no detectable nuclease contamination in purified PACT as measured by the integrity of both dsRNA/ssRNA co-incubated with PACT for 2 h at 37 °C (Supplementary Fig. 8D).

To explore the mechanism of inhibition, we first considered whether PACT might constitutively bind to and inhibit PKR independently of dsRNA. However, PACT's inhibition varied strongly with dsRNA length and concentration (Fig. 4B). For instance, at 50 nM PACT, PKR activity with 112 bp dsRNA was nearly completely abrogated, whereas activity with 42 or 62 bp dsRNA was minimally affected. Additionally, inhibition was more efficient at lower dsRNA

concentrations (Fig. 4C). These results support the idea that PACT's inhibition, at physiological concentration, may occur on dsRNA.

We also considered whether PACT might sequester dsRNA away from PKR. If this were the case, its impact would not become apparent until PACT concentrations were high enough to stably occupy a significant portion of the dsRNA. However, even at 50 nM PACT--where less than 10% of PKR-binding sites are stably occupied by PACT (Fig. 2A)--PKR activity with 112 bp dsRNA (0.25 ng/μl) decreased by more than 90% (Fig. 4C), suggesting that PACT's inhibition of PKR activity does not require dsRNA sequestration, consistent with our in vivo observation in Fig. 2.

Finally, we investigated whether PACT interferes with PKR's motion on dsRNA, potentially disrupting its self-interactions and subsequent autophosphorylation. FRET experiments using Cy5-

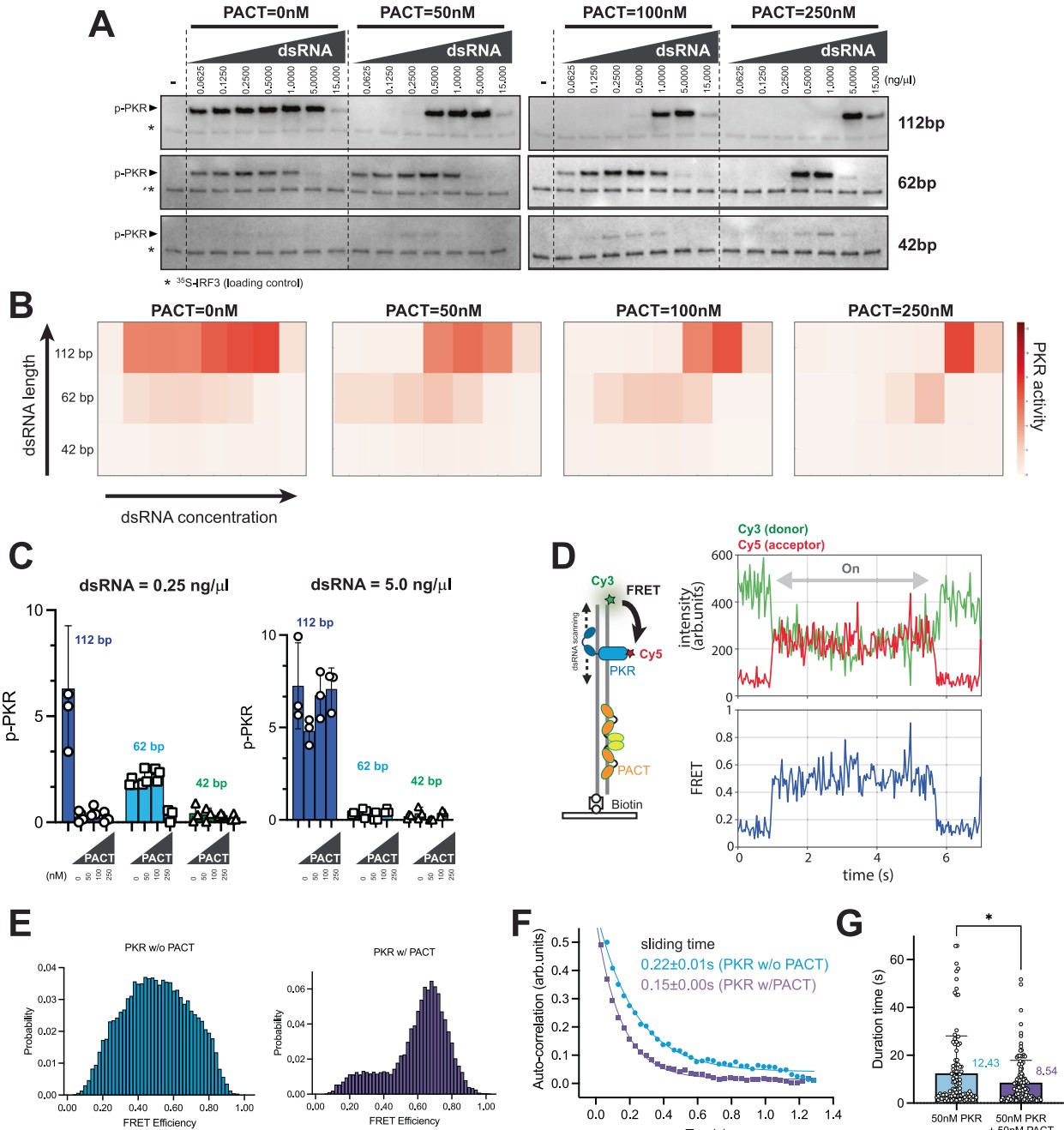

**Fig. 4 | PACT inhibits PKR by restricting its motion on dsRNA, leading to length-dependent inhibition. A** In vitro PKR kinase assay as in Fig. 3B, with increasing PACT concentrations (0–250 nM). [³⁵S]-IRF3 served as a loading control (*). Results were reproduced in 3 replicates. **B** Heatmap of relative PKR activity from gels in **A**, representing means of 3 biological repeats. **C** Relative PKR activity with 0.25 ng/µl (left) and 5 ng/µl (right) of 112, 62, and 42 bp dsRNA, with increasing PACT concentrations (0–250 nM). Mean ( ± SD) of 3 biological repeats. **D** Schematic of single-molecule FRET experiments monitoring PKR diffusion on 112 bp dsRNA in the presence of PACT. Cy5-labeled PKR (acceptor, 50 nM) was mixed with unlabeled PACT (50 nM) and added to a chamber with immobilized Cy3-labeled dsRNA (donor). Cy3 and Cy5 fluorescence and Cy3-Cy5 FRET were monitored upon Cy3 excitation using Single-molecule TIRF (sm-TIRF) microscopy. PKR K296R was used to ensure analysis of PKR movement before its activation. Created in BioRender. torres, c. (2025) https://BioRender.com/g96x368. **E** FRET histograms for PKR

without PACT (blue) and with PACT (purple). To compare the range of PKR motions with and without PACT, we focused our analysis on traces showing a high FRET state ( > 0.8). Without PACT, PKR exhibits diffusion across a broad FRET range (0.2–0.8), while its movement is more restricted with PACT, showing FRET values between 0.6 and 0.8. See "Methods" for details on histogram quantification. **F** Autocorrelation analysis of FRET signal for PKR without PACT (blue) and with PACT (purple). Scanning times were calculated by single exponential curve fitting (see "Methods"). 236 and 93 diffusion events were analyzed for PKR without and with PACT, respectively. **G** Residence time of PKR on dsRNA ("on" time) in the presence and absence of PACT (50 nM). Average values (in seconds) are shown on the right. 236 and 117 diffusion events were analyzed for PKR without and with PACT, respectively. Data represent mean values ± SD. *P* values were calculated by a paired two-tailed *t*-test. (**P* < 0.05). Source data and exact *P* values are provided as a Source Data file.

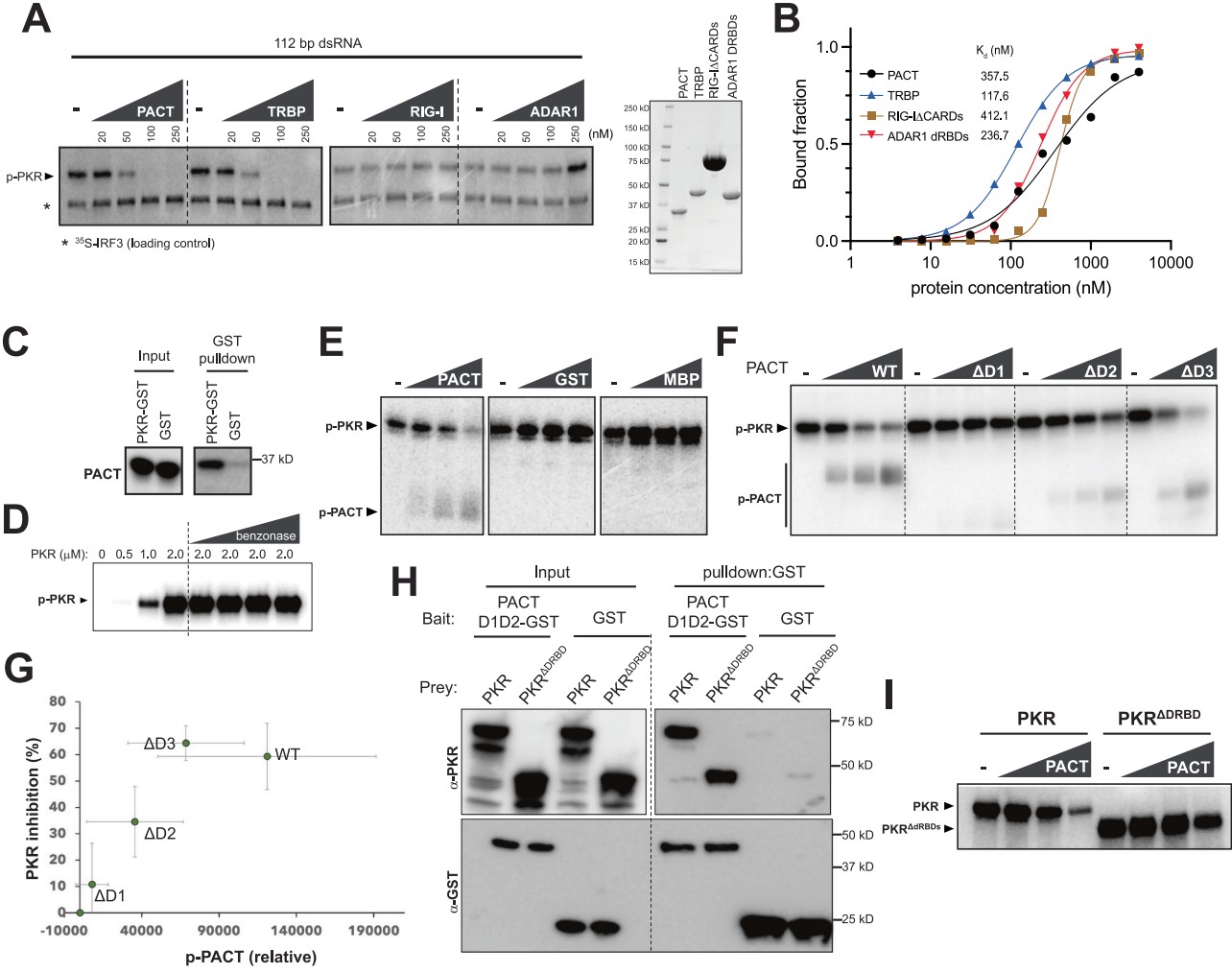

**Fig. 5 | PACT's inhibition of PKR involves weak but direct and specific protein-protein interactions. A** In vitro PKR kinase assay using 112 bp dsRNA (0.25 ng/μl) in the presence of 0, 20, 50, 100, 250 nM PACT, TRBP, RIG-I ΔCARDs, ADAR1 DRBDs. Right: SDS-PAGE showing purity of the proteins. The results were reproduced in 3 independent replicates. **B** dsRNA binding curve of PACT, TRBP, RIG-I ΔCARDs, ADAR1 DRBDs, derived from the native gel-shift assay in Supplementary Fig. 10A monitoring 112 bp dsRNA binding. For curve fitting, Hill equation was used with the assumption of 8 protein binding sites per 112 bp dsRNA (see Methods). **C** Pulldown of PACT using PKR-GST or GST. All proteins were purified and treated with benzonase to remove potential nucleic acid contaminants. Catalytically dead K296R PKR was used. The results were reproduced in 3 independent replicates. **D** In vitro PKR kinase assay using an increasing concentration of PKR in the absence of dsRNA. Benzonase was added during the reaction (in addition to during purification) to

ensure that the observed PKR activity is RNA-independent. The results were reproduced in 2 independent replicates. **E** RNA-independent PKR kinase assay with 1 μM PKR in the presence of 0, 1.25, 2.5, 5 μM PACT, GST, MBP. The results were reproduced in 3 independent replicates. **F** RNA-independent PKR kinase assay with 1 μM PKR in the presence of 0, 1.25, 2.5, 5 μM PACT wild type, ΔD1, ΔD2; and 0, 2.5, 5 μM ΔD3. **G** Correlation between the levels of PKR inhibition and PACT phosphorylation from **F**. Values are mean (± SD) of 2 biological replicates. **H** Pulldown of PKR or PKRΔDRBDs using PACT D1D2-GST or GST. Catalytically dead K296R mutant was used for both PKR and PKRΔDRBDs. The results were reproduced in 3 independent replicates. **I** RNA-independent PKR kinase assay with 1 μM PKR or 2 μM PKRΔDRBDs in the presence of 0, 1.25, 2.5, 5 μM PACT. Higher concentration of PKRΔDRBDs was due to its lower activity. The results were reproduced in 3 independent replicates. Source data are provided as a Source Data file.

labeled PKR revealed that in the presence of 50 nM PACT (unlabeled), PKR still bound dsRNA and showed rapid FRET fluctuations (Fig. 4D and Supplementary Fig. 9A, B), indicating that PKR binding was not blocked by PACT. However, we found a reduced range of FRET fluctuations in the presence of PACT (Fig. 4D and Supplementary Fig. 9A, B), suggesting a diminished scanning range. For quantitative comparison, we focused on binding events containing the high FRET state (FRET > 0.8; where PKR comes in close proximity to the labeled dsRNA end) and generated FRET histograms. In the absence of PACT, PKR exhibited diffusion across a broad FRET range (0.2 to 0.8), indicating wide movement (Fig. 4E). In contrast, with PACT present, the movement was confined to FRET states between 0.6 and 0.8 (Fig. 4E). Autocorrelation analysis further revealed that PACT reduced PKR's sliding time, consistent with restricted scanning motion (Fig. 4F).

Finally, the residence time of PKR on dsRNA was also shortened (Fig. 4G). These observations suggest that PACT interferes with PKR diffusion, which likely explains the observed inhibitory activity

## PACT's inhibition of PKR involves weak but direct protein-protein interactions

To understand PACT's mechanism further, we asked whether a mere dsRNA-binding activity is sufficient to inhibit PKR. We compared PACT with other dsRNA-binding proteins. They were PACT's closest paralog TRBP, and dsRNA-binding regions of RIG-I (ΔCARD) and ADAR1 (DRBD1, 2, 3). The results showed that TRBP inhibited PKR similar to PACT, but RIG-I and ADAR1 did not under the equivalent condition (Fig. 5A). This is despite the fact that RIG-I and ADAR1 bind dsRNA with comparable affinities (Fig. 5B and Supplementary Fig. 10A). These

observations suggest that a mere dsRNA binding activity of PACT may be insufficient for PKR inhibition, implicating a direct and specific interaction with PKR. Consistent with this prediction, we observed a direct interaction between purified PACT and GST-PKR in the absence of RNA, as measured by GST pull-down (Fig. 5C). The affinity appeared low as co-purified PACT could not be detected by Coomassie or Krypton staining, requiring more sensitive western blotting. We also observed their interaction using an orthogonal approach involving microscale thermophoresis, but their affinity in the absence of dsRNA was again too low to be accurately determined (Supplementary Fig. 10B).

We hypothesized that the PACT-PKR's direct interaction may be facilitated by their co-occupancy on dsRNA. Such RNA-facilitated protein-protein interaction is ubiquitous and can be recapitulated in the absence of RNA at high protein concentration, which bypasses the RNA requirement. Thus, we tested whether PACT could inhibit PKR activity independent of RNA at high protein concentrations. This hypothesis was testable because PKR activation can also bypass the dsRNA requirement at high PKR concentration (Fig. 5D). We confirmed that the PKR activity at high protein concentration was not due to potential contaminating nucleic acids, as the addition of increasing amounts of benzonase had little impact on PKR under these conditions (Fig. 5D). The addition of $1-5\,\mu M$ PACT inhibited PKR in a dose-dependent manner (Fig. 5E), suggesting that PACT can inhibit PKR independent of dsRNA at high, albeit non-physiological concentrations of both PKR and PACT. Unrelated proteins, such as GST and MBP, did not show a similar PKR-inhibitory activity at equivalent concentrations (Fig. 5E). Notably, PACT's inhibition of PKR was accompanied by PACT's phosphorylation by PKR. No such phosphorylation was observed with GST and MBP, despite having multiple surface-exposed Ser/Thr residues (9 and 20 for GST and MBP). Since phosphorylation requires direct interaction between a kinase and its substrate, this result further supports the notion that PACT forms a specific interaction with PKR.

We next examined which domains of PACT are involved in PKR suppression. In order to uncouple the requirements for dsRNA binding and PKR suppression, we continued using the assay condition with high PKR and PACT concentrations in the absence of dsRNA. Deletion of DRBD1 or DRBD2 of PACT, which are important for dsRNA binding (Supplementary Fig. 10C), had a significant impact on PACT's PKR-suppressive activity (Fig. 5F), whereas deletion of DRBD3 of PACT had a less significant impact on either dsRNA binding or PKR suppression (Fig. 5F and Supplementary Fig. 10C).

Interestingly, we observed an inverse correlation between the levels of p-PKR and p-PACT (Fig. 5F). In other words, the more effectively a PACT variant inhibits PKR, the more susceptible that variant is to being phosphorylated by PKR (Fig. 5G). This is the opposite of what would be expected if phosphorylation were non-specific. Furthermore, the varying levels of phosphorylation by PKR did not simply correlate with the number of surface-exposed Ser/Thr residues (43, 28, 34, and 25 for PACT, ΔDRBD1, ΔDRBD2, and ΔDRBD3, respectively), further arguing against non-specific phosphorylation. Instead, the observed correlation between PKR-mediated phosphorylation and PACT's ability to inhibit PKR suggests a specific interaction between the two proteins that is important for PACT's inhibitory mechanism.

Given the importance of DRBD1 and DRBD2 in PKR inhibition, and the less significant role of DRBD3, we asked whether DRBD1-2 of PACT can interact with PKR. Consistent with this notion, PKR was pulled down by PACT DRBD1-2 in the absence of RNA (Fig. 5H). PACT also interacted with PKRΔDRBD (Fig. 5H) and inhibited the kinase activity of PKRΔDRBD (Fig. 5I), albeit not to the same extent as full-length PKR. These results together suggest that PACT can directly interact with PKR for the inhibition, and this interaction involves PACT DRBD1-2 and PKR kinase domain.

## PACT DRBD2 utilizes a surface distinct from the dsRNA interface to inhibit PKR

Our effort to further characterize the interaction between PKR and PACT using structural biology approaches proved challenging, likely due to the weak nature of binding and/or heterogeneous conformational states. Alphafold predictions were also unsuccessful. Therefore, we explored the interaction using a mutagenesis approach. To narrow down the potential interface, we surveyed other DRBDs known to form stable complexes with various protein partners and with known complex structures. These included TRBP DRBD3 in complex with Dicer[39], TRBP DRBD3 in complex with RPAP3[41], and Staufen DRBD5 in complex with Miranda[40]. We also determined the NMR structure of PACT DRBD3 homodimer, which revealed that two DRBD3 join their β-sheets (β1-β2-β3) through a parallel β3:β3 interaction (Fig. 6A inset, Supplementary Fig. 10D, and Supplementary Table 1).

Comparison of these structures showed a striking commonality among all of these interactions––they all utilize the same face of the DRBD opposite from the surface that is equivalent to the dsRNA interface in the canonical DRBD (Fig. 6A). In particular, all utilize β3 as a key component in the protein interactions (Fig. 6A). Based on these observations, we asked whether PACT also inhibits PKR through β3 of DRBD1 and DRBD2.

We generated PACT variants harboring mutations in and around β3 of DRBD1 (T76R, C77R) and DRBD2 (K173E, K187E, K191E) (Fig. 6B). All mutants showed comparable affinities for dsRNA as WT PACT (Fig. 6B and Supplementary Fig. 10E), suggesting that they are folded correctly. This is also consistent with the structure that these residues are far from the dsRNA interface. All three DRBD2 mutations, however, impaired PKR-inhibitory activity of PACT (Fig. 6C). The DRBD1 mutations did not affect the PKR-inhibitory function of PACT, suggesting different surface utilization (see the discussion in Supplementary Figs. 10F, G). We again observed the correlation between PKR's ability to phosphorylate PACT and PACT's inhibitory effect on PKR (Fig. 6D), further supporting the notion that PACT exerts PKR inhibition through direct protein-protein interaction. Finally, we found that the DRBD2 mutants K187E and K191E were defective in restoring cell viability and inhibiting the PKR activity in PACT-dependent cells in our complementation assay (Fig. 6E and Supplementary Fig. 11A, B). In contrast, C77R was comparable to WT PACT in both the in vitro PKR activity assay and cellular assays (Fig. 6C, E and Supplementary Fig. 11A). Interestingly, K173E, which showed an impaired PKR-inhibitory effect in vitro, did not exhibit a significant defect in the cellular assays (Fig. 6C, E and Supplementary Fig. 11A), suggesting a potential compensatory mechanism in cells. Nevertheless, our mutagenesis data on K187E and K191E support the notion that PACT inhibits PKR through a direct protein-protein interaction, involving the PACT DRBD2 surface analogous to other DRBD-protein interfaces.

## Discussion

PKR is a conserved sensor for foreign dsRNA responsible for activating the integrated stress response during viral infection and contributing to the pathogenesis of a broad range of immune diseases and neurodysregulation[11-15]. Despite its importance, the mechanisms regulating PKR to prevent aberrant activation by endogenous dsRNA are only beginning to be understood. Through mechanistic studies, we here demonstrate that PACT, whose functions have been controversial, plays an essential role in restraining PKR and preventing its inappropriate activation by endogenous dsRNA formed by Alu elements. Our biochemical analyzes suggest that previous reports of PACT-dependent activation of PKR[29-32] may have resulted from incomplete RNA removal from PACT, which we found requires extensive nuclease treatment and elaborate purification steps.

Our data suggest that PACT inhibits PKR through an unusual mechanism dependent on dsRNA length and concentration (Fig. 7A). Although PACT and PKR share the endogenous dsRNA ligands,

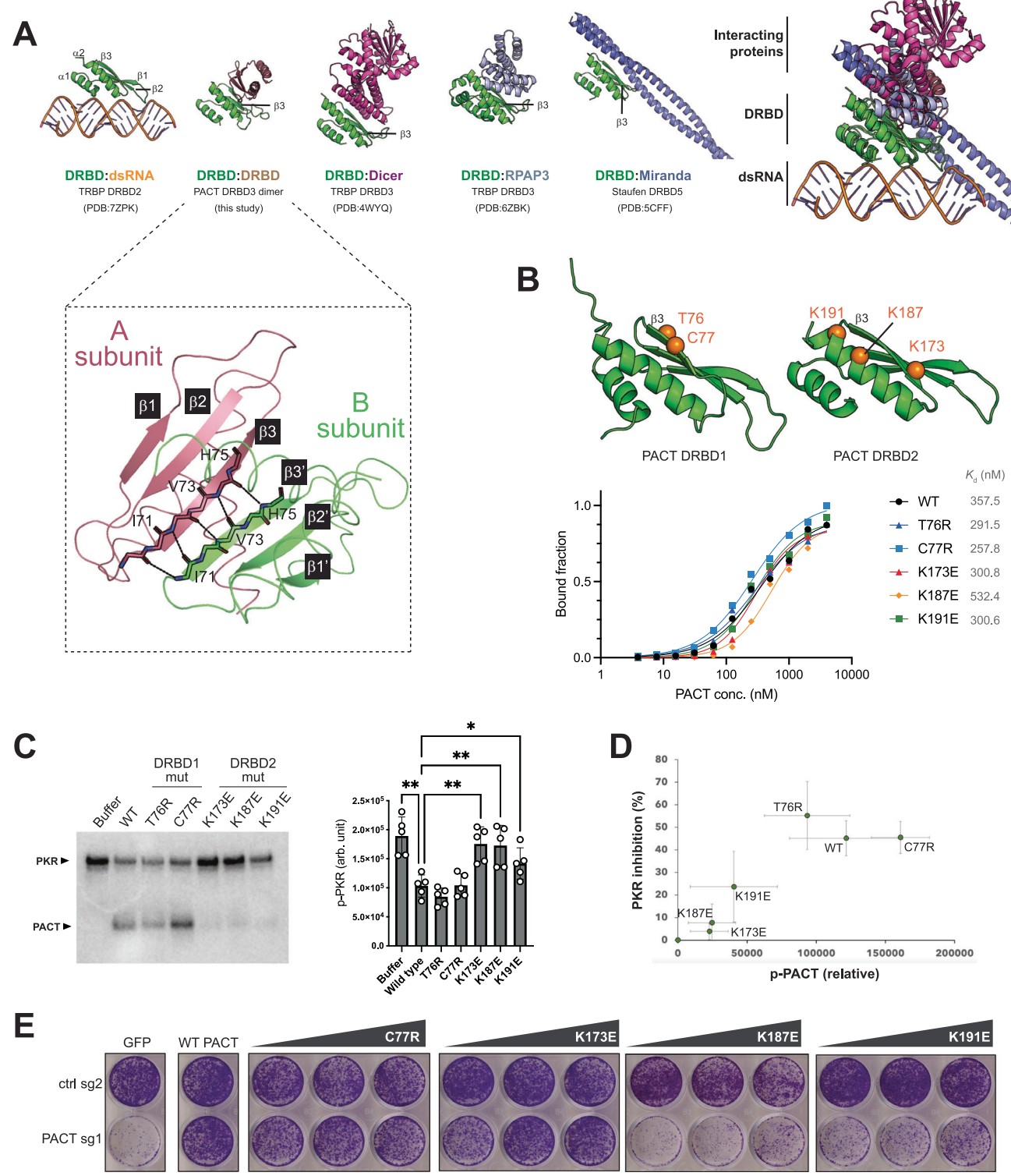

primarily IR-Alus, PACT does not block PKR binding to these dsRNAs per se. Instead, a substoichiometric amount of PACT binding to dsRNA is sufficient to inhibit the bulk of PKR activity. Single molecule analyzes suggest that both PKR and PACT scan along dsRNAs––a characteristic common to proteins with tandem repeats of nucleic acid-binding domains[53,54,56]. PKR utilizes its scanning ability to encounter other PKR molecules for autophosphorylation, explaining its preference for long dsRNAs (Fig. 7B). Conversely, PACT scans dsRNA to interact with PKR, restricting PKR's movement and promoting its dissociation from dsRNA. This decreases the likelihood of PKR molecules encountering

one another and undergoing autophosphorylation, thereby inhibiting PKR without sequestering dsRNA (Fig. 7B). Consequently, PACT's inhibition of PKR is more robust when dsRNA is longer and less abundant (Fig. 7A)––conditions that favor co-occupancy of PACT and PKR on the same RNA. The net result of such activity is to selectively raise the tolerance threshold for long, more potent dsRNAs for PKR without compromising its inherent activity (Fig. 7A).

Our results also suggest that PACT's inhibition involves more than simple dsRNA binding; it likely includes a direct interaction between PACT and the kinase domain of PKR. This interaction appeared weak in

**Fig. 6 | PACT DRBD2 utilizes the surface distinct from the dsRNA interface to inhibit PKR. A** Structures of DRBDs in complex with dsRNA or protein partners, viewed from the same perspective relative to the DRBDs. Structure of PACT DRBD3 dimer is from this study. All other structures were previously published. The comparison shows the conserved protein:protein interface involving β3 away from the dsRNA binding surface. Inset: NMR structure of PACT DRBD3 showing its homo-dimerization by joining their β-sheets (β1- β2- β3) through a parallel β3: β3 interaction. **B** Sites of mutations on the putative protein:protein interface of PACT DRBD1 and DRBD2. Bottom: dsRNA binding curves for PACT wild type, DRBD1 mutants T76R, C77R, and DRBD2 mutants K173E, K187E, K191E, derived from EMSA in Supplementary Fig. 10E. **C** RNA-independent PKR kinase assay with 1 µM PKR in

the presence of 5 µM PACT wild type, T76R, C77R, K173E, K187E, K191E. Right: quantification from 5 independent experiments. Data are presented as mean values ± SD. *P*-values were calculated by one-way ANOVA followed by Tukey multiple comparisons test. \*\**p* ≤ 0.0021, \**p* ≤ 0.0332; (ns) *p* > 0.12. *P* values: (buffer vs. wild type = 0.0019, wild type vs. K173E = 0.0034, wild type vs. K187E = 0.0093, wild type vs. K191E = 0.0231). **D** Correlation between the levels of PKR inhibition and PACT phosphorylation from **C**. Values are mean (± SD) of 5 biological repeats. **E** Crystal violet staining assay showing cell viability in NCI-H727 cells after control gene-KO (ctrl) and PACT-KO complemented with expression of PACT wild type or increasing amounts of C77R, K173E, K187E, and K191E. Source data are provided as a Source Data file.

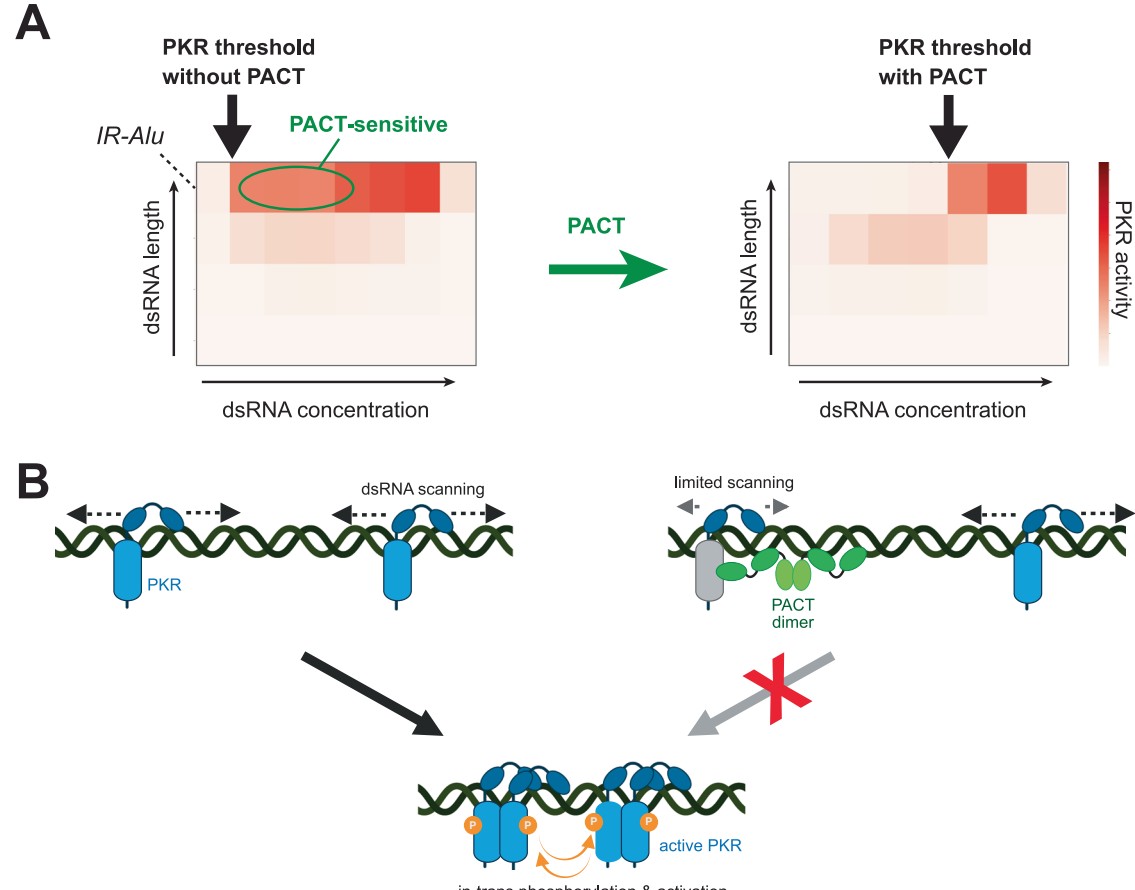

**Fig. 7 | Model for PKR activation and PACT-mediated inhibition. A** Heatmaps showing dsRNA length and concentration dependance for PKR activity in the absence and presence of PACT. PACT restricts PKR selectively within a specific range of the dsRNA length-concentration space, resulting in the selective elevation of PKR's activity threshold for longer dsRNAs (such as IR-Alus). Notably, PACT has minimal impact on PKR when stimulated with high concentrations of long dsRNA, suggesting that PACT acts as a homeostatic regulator without compromising PKR's inherent activity. **B** Model of PKR activation by dsRNA and its inhibition by PACT. PKR activation involves dsRNA scanning, molecular collisions, and subsequent

trans-autophosphorylation on long dsRNA. PACT also scans along dsRNA, restricting PKR's movement, preventing molecular collision between PKR molecules. This inhibition requires more than PACT's dsRNA-binding activity; it also involves a direct, albeit weak, interaction with PKR's kinase domain, facilitated when both PKR and PACT are on the same dsRNA. As a result, PACT's inhibition of PKR is more robust when dsRNA is longer and less abundant––conditions that promote co-occupancy––explaining how PACT modulates its inhibitory function based on dsRNA length and concentration. Created in BioRender. torres, c. (2025) https://BioRender.com/g96x368.

the absence of dsRNA, and would likely occur only when both PKR and PACT are bound to the same dsRNA. Although the low affinity of this interaction has made detailed characterization difficult, the phosphorylation of PACT by PKR, which requires direct contact, supports this model. Moreover, mutations in PACT at putative interface regions, away from the dsRNA-binding domains, affect PKR activity both in vitro and in cells. These interactions may help explain why PACT specifically modulates PKR function with minimal effect on other dsRNA-

binding proteins, such as ADAR1, OASes, and RLRs, though alternative explanations cannot be ruled out. Low-affinity interactions like these, which are often challenging to study, may be more common in nature than appreciated and could play important roles in mediating complex biological processes.

In summary, PACT's ability to inhibit PKR in a dsRNA length- and concentration-dependent manner contrasts with other PKR inhibitors, such as viral inhibitors (e.g., vaccinia viral protein K3L), which use RNA-

independent high-affinity interactions to globally inhibit PKR[57,58]. Our work thus reveals a mechanism by which PKR and PACT function together to maintain immune homeostasis amid fluctuating levels of endogenous dsRNA.

# Methods

## Material preparation

**Cell lines and culture conditions**. All cancer cell lines were grown and maintained in RPMI media supplemented with 10% fetal bovine serum (FBS), and 1% penicillin, streptomycin, and L-glutamine, except for NCI-H2023, which was maintained in DMEM/F12 media. The following cancer cell lines were obtained from the American Type Culture Collection (ATCC): ZR-75-1, HCC1428, NCI-H1437, NCI-H2023, NCI-H1650, HCC1806, MDA-MB-157, NCI-H2286, NCI-H727, and SW1271. Prior to shipping each cell line, ATCC performs cell line authentication. All cell lines were tested for mycoplasma regularly.

**Plasmids**. Bacterial expression plasmids for PACT and PKR were made in pET47b and pET29b, respectively. PACT wild type was inserted between BamHI and SalI sites. PKR was inserted between NdeI and KpnI sites. Point mutants and truncation mutants for PACT: ΔD1 (residues 124–313), ΔD2 (Δresidues 110–198), ΔD3 (residues 1–196), D3 (residues 235–313), and for PKR: ΔDRBDs (residues 170–551) and all point mutations were generated by site-directed mutagenesis. PACT DRBD3 (residues 235-313) was cloned in pET47b between BamHI and SalI sites. TRBP wild type was inserted in pET28a between NheI and HindIII sites, and ADAR1 DRBDs (residues 496–800) were inserted in pET47b between BamHI and HindIII sites. RIG-I ΔCARDs and MDA5ΔCARDs were used from a previous study[48,59]. For cellular studies, entry clones for *PACT* (wildtype, C77R, K173E, K187E, K191E, and 4KE) and *PKR K296R* were generated through PCR-amplification of the respective *PACT* and *PKR K296R* open reading frames (ORF) described above. These entry clones were sub-cloned into a Gateway donor vector. Overlap PCR was performed to introduce (1) silent mutations into the protospacer adjacent motif (PAM) sequences targeted by PACT sgRNA 1 for *PACT* clones or PKR sgRNA 1 for *PKR K296R* to render these constructs resistant to CRISPR-Cas9 editing by these sgRNAs; and (2) FLAG tags to the C terminus of *PACT* clones or a hemagglutinin tag to the C terminus of *PKR K296R*. The individual *PACT* constructs and a *GFP* control construct were sub-cloned into the pLEX306 lentiviral expression vector (Addgene) under the control of a human PGK promoter. The *PKR K296R* construct was sub-cloned into the pLEX307 lentiviral expression vector (Addgene) under the control of an EF-1α promoter. Each expression vector was then transfected into HEK 293T cells to generate lentivirus. Lentiviral transduction of target cell lines was performed as described below. For the *PACT* clones (wild-type, C77R, K173E, K187E, K191E, and 4KE), increasing volumes of lentivirus were used to transduce target cell lines to achieve a gradient of PACT protein expression in target cell lines.

**Proteins**. All PACT constructs (except 4KE, D3), TRBP, and ADAR1 DRBDs (residue 496-800) were expressed and purified from *E. coli* BL21(DE3) using a combination of Ni-NTA, heparin affinity, and size-exclusion chromatography. Cells were lyzed using high-pressure homogenization with Emulsiflex C3 (Avestin) and centrifuged. The supernatant was loaded onto Ni-NTA agarose beads and washed with 50 column volumes (CV) of high salt buffer (2 M NaCl) to get rid of bound RNA contaminants before elution in 50 mM sodium phosphate pH 7.5, 1 M NaCl, 10% glycerol, 300 mM imidazole. The eluate was further treated with HRV-3C protease and benzonase overnight to remove the His-tag and contaminating RNAs, respectively. After tag removal, the protein was subjected to heparin affinity chromatography followed by size-exclusion chromatography (SEC) in 50 mM HEPES pH 7.5, 250 mM NaCl, and 2 mM DTT to further purify proteins and separate from benzonase. Removal of benzonase was confirmed by the stability of

dsRNA co-incubated with purified protein at 37 °C for 2 h. For 4KE and D3 mutants, benzonase treatment and heparin affinity chromatography steps were skipped since they were RNA-binding defective.

PKR constructs were co-expressed and purified from *E. coli* Rosetta2 (DE3) (Novagen) using a combination of Ni-NTA and size-exclusion chromatography. NiNTA was done in the same buffer as PACT, except the wash was done with 100 CV high salt buffer 50 mM Tris 8.0, 1 M NaCl, 5 mM β-mercaptoethanol (BME). The eluate from NiNTA was subjected to SEC in a 16/600 Superdex 200 column in 50 mM Tris pH 7.5, 300 mM NaCl, 5 mM BME. The eluate was sequentially treated with lambda protein phosphatase (PPase) (NEB) and benzonase (EMD Millipore) in the presence of 1 mM MnCl$_2$ and 1 mM MgCl$_2$, respectively. PKR was subjected to another round of SEC in 50 mM Tris pH 7.5, 100 mM NaCl, and 5 mM BME to get rid of any contaminant PPase or benzonase. Note that PKR wild-type constructs were co-expressed with PPase-expressing pET21b for purification. For K296R mutants, both PPase co-expression and post-SEC PPase treatment were skipped since the mutation is catalytically inactive. RIG-I ΔCARDs and MDA5 ΔCARDs were purified as reported previously[48,59].

**RNAs**. All dsRNAs used in this study were generated by in vitro T7 transcription as reported previously[51]. The templates for RNA synthesis were made by PCR amplification. The sequences of 42, 62, 112, and 512 bp dsRNAs were taken from the first 30, 50, 100, and 500 bp of the MDA5 gene, respectively, flanked by 5′-gggaga and 5′-tctccc (See Supplementary Table 3). The two complementary RNA strands were co-transcribed, and the duplex was purified using 8.5% acrylamide gel electrophoresis. RNA was gel-extracted using the Elutrap electroelution kit (Whatman), ethanol precipitated, and stored in 20 mM HEPES pH 7.0. For 32P-labeled RNA, in vitro transcription was done in the presence of trace amounts of [α-32P]-UTP, followed by RNA clean-up using RNA Clean & Concentrator Kit (Zymo Research). The RNA concentration was estimated using Qubit Fluorometers (Thermo).

## CRISPR-Cas9 gene knockout

Single guide RNA sequences were designed using the sgRNA Designer tool (The Broad Institute Genomics Perturbation Platform) (http://portals.broadinstitute.org/gpp/public/analysis-tools/sgrna-design). sgRNA sequences are displayed in Supplementary Table 2. sgRNAs were cloned into the Cas9-expressing lentiviral vector lentiCRISPRv2 (Addgene) or a modified lentiCRISPRv2 construct that expresses two different sgRNAs under the control of separate human and mouse U6 promoters. Individual lentiCRISPRv2 vectors were introduced along with packaging vectors into HEK 293T cells via calcium phosphate transfection according to the manufacturer's protocol (Clontech). Lentivirus was harvested at 48 and 72 h after transfection in RPMI media supplemented with 10% FBS and filtered with 45 μm filters before transduction of target cancer cell lines using centrifugation at 1000 × *g* for 2 h in the presence of 8 mg/mL of polybrene (Santa Cruz Biotechnology). Transduced cell lines were selected in 2 μg/mL puromycin and/or 10 μg/mL blasticidin for at least 5 days prior to use in assays. Protein lysates were collected from the transduced cells, and protein levels of the targeted gene(s) were assessed by immunoblotting.

## Cell viability assays

Cell counting was performed using a Coulter Particle Counter (Beckman-Coulter). For ATP bioluminescence experiments, cells were plated at a density of 3000 cells per well in 96-well assay plates (Corning). ATP bioluminescence was assessed at 13 days after gene knockout with the CellTiter-Glo Luminescent Cell Viability Assay (Promega). For crystal violet staining assays, cells were plated at a density of 10,000 to 20,000 cells per well in 12-well tissue culture plates. Once the control cells grew to near confluency, each well was washed twice with ice-cold

PBS, fixed on ice with ice-cold methanol for 10 min, stained with 0.5% crystal violet solution (made with 25% methanol) for 10 min at room temperature, and washed at least four times with water. All cell viability assays were performed in triplicate.

## Antibodies and immunoblotting
Cells were lysed in RIPA lysis buffer (Thermo Fisher Scientific) supplemented with 1× protease and phosphatase inhibitor cocktails (Roche). Protein concentrations were obtained using the BCA Protein Assay Kit (Pierce), and 6X Laemmli SDS sample buffer (Thermo Fisher Scientific) was added to protein extracts. Protein concentrations were normalized between all samples within an experiment and boiled above 95 °C for 10 min. Proteins were resolved on 4–12% Bis-Tris gradient gels, transferred to nitrocellulose membranes, and immunoblotting with primary and secondary antibodies was performed according to standard procedures. All primary antibodies were used at a dilution of 1:1000 except anti-β-actin and anti-vinculin, which were used at a dilution of 1:10,000. Secondary antibodies from LI-COR Biosciences were used at a dilution of 1:10,000. Selected immunoblots were stripped with Restore PLUS Western Blot Stripping buffer (Thermo Fisher Scientific) prior to repeat immunoblotting. Immunoblots were imaged using the LI-COR digital imaging system. All immunoblots were cropped to optimize clarity and presentation.

## In vitro PKR kinase assay
For RNA-dependent PKR kinase assays, 100 nM PKR was mixed with the mentioned concentration of RNA, 2 mM ATP and 1 μCi [γ$^{32}$P]-ATP in 20 mM HEPES pH 7.5, 100 mM NaCl, 1.5 mM MgCl$_2$, 2 mM DTT in the presence of mentioned concentration of PACT and the reaction incubated at 30 °C for 30 min. For RNA-dependent PKR kinase assays, 1 to 1.5 μM PKR was first pre-treated with Benzonase (Millipore) in the presence of 2 mM MgCl$_2$ at room temperature for 15 min. For the experiments involving PACT, PACT was also pre-treated with Benzonase (Millipore) in the presence of 2 mM MgCl$_2$ at room temperature for 15 min. Both PACT and PKR, after benzonase treatment were mixed together and incubated at room temperature for 15 min before adding 2 mM ATP and 1μCi [γ$^{32}$P]-ATP in the final buffer comprising of 20 mM HEPES pH 7.5, 150 mM NaCl, 1.5 mM MgCl$_2$, 2 mM DTT and incubating at 37 °C for 10 min. After the reaction incubation time, the samples were quenched in SDS gel loading buffer and subjected to SDS-PAGE. The gels were exposed to an imaging plate from Cytiva, and Phosphor imaging was done using an Amersham Typhoon Imager (GE). Quantification was done using ImageJ.

## fCLIP-seq
The procedure was adopted from the previous study[3]. Cells were crosslinked with 0.1% paraformaldehyde for 10 min at RT followed by quenching with 150 mM glycine for 10 min at RT. The crosslinked cells were then resuspended in 20 mM Tris pH 7.5, 15 mM NaCl, 10 mM EDTA, 0.5% NP-40, 0.1% Triton X-100, 0.1% sodium deoxycholate, 1x mammalian protease inhibitor cocktail (G Biosciences), 40 U/ml RNase inhibitor (NEB) and 40U/ml DNase I (NEB) and incubated on ice for 10 min. The cells were lysed by sonication using Covaris E220 focused-ultrasonicator (PIP: 140, DF: 5, Cycles/Burst: 200, Time: 5 min). The lysate was centrifuged at 18,000 g for 30 min, and the supernatant was added to pre-equilibrated protein G Dyanbeads (Invitrogen) along with Anti-HA (CST #3724), Anti-PKR (CST #12297), or Anti-Flag (Millipore Sigma #F1804) antibody. The mixture was incubated at 4 °C for 2 h before washing beads with 0.1% SDS, 0.5% NP-40, 0.5% sodium deoxycholate in 1X PBS and eluting in 200 mM Tris-HCl pH 7.5, 100 mM NaCl, 20 mM EDTA, 2% SDS, and 7 M Urea with constant shaking at 950 rpm at 25 °C for 1-2 h. The eluted protein:RNA complex was then de-crosslinked overnight at 65 °C with constant shaking at 1000 rpm in the presence of 2.7 U/ml proteinase K (NEB) to digest the proteins. The RNA was then extracted by phenol:chloroform extraction followed by

isopropanol precipitation. The extracted RNA was further treated with DNase I (NEB) for 30 min at 37 °C to completely get rid of any DNA contamination. The purified RNA was used for cDNA library preparation using SMARTer Stranded Total RNA-Seq Kit v3 - Pico Input Mammalian according to the manufacturer's instructions. The cDNA library was sequenced using the Illumina Partiallane NovaSeq platform.

## RT-qPCR
Total RNAs were extracted using Directzol RNA miniprep kit (Zymo research) and cDNA was synthesized using High-Capacity cDNA reverse transcription kit (Applied Biosystems) according to the manufacture's instructions. Real-time PCR was performed using a set of gene-specific primers, SYBR Green Master Mix (Applied Biosystems), and the CFX96 Real-Time PCR Systems (Biorad).

## Native gel-shift assay
For quantitative analysis, 0.2 ng/μl $^{32}$P labeled 112 bp was incubated with 0, 3.9, 7.8, 15.6, 31.3, 62.5, 125, 250, 500, 1000, 2000, 4000 nM PACT proteins or TRBP, RIG-I ΔCARDs, ADAR1 DRBDs in 20 mM HEPES pH 7.5, 150 mM NaCl, 2 mM DTT and incubated at RT for 15 min. The samples were then subjected to Bis-Tris native PAGE (Life Technologies). The gels were exposed to an imaging plate from Cytiva, and Phosphor imaging was done using Amersham Typhoon Imager (GE), followed by quantification in ImageJ. The quantification was done as previously reported[51]. Briefly, the bound fraction was calculated using the equation:

$$\text{Bound fraction} = \sum_{n=0}^{8} I(n) * n \Big/ \sum_{n=0}^{8} I(n) * 8 \qquad (1)$$

where I(n) refers to the intensity of the nth complex band. Free dsRNA is referred to by the 0th complex band. The underlying assumption is that 112 bp dsRNA has eight binding sites, and the nth complex band corresponds to dsRNA with n sites occupied.

For non-quantitative analysis, 25 ng/μl RNA (42 or 62 bp) was incubated with 0, 78, 156, 312.5, 625, 1250, 2500 nM nM PKR at RT for 15 min before being subjected to native PAGE. The gel was stained with Sybr Gold (Thermo) and imaged using iBright FL1000 (Invitrogen).

## Multi-Angle Light Scattering (MALS)
PKR K296R mutant (0.7 mg/ml) was mixed with 50 ng/μl RNA (42 or 62 bp) in 20 mM HEPES pH 7.5, 150 mM NaCl, 2 mM DTT, and incubated at RT for 30 min. The mix was then filtered using a 0.22 μm filter and then loaded onto a Superdex 200 Increase 10/300 column (Cytiva) attached to MiniDAWN detector (Wyatt Technology) in 20 mM HEPES pH 7.5, 150 mM NaCl, 2 mM DTT. The data were analyzed using ASTRA7.3.1 software (Wyatt Technology).

## Electron microscopy
PKR, PACT or MDA5ΔCARDs (0.3 μM) was incubated with 512 bp dsRNA (0.6 ng/μl) in 20 mM HEPES pH 7.5, 150 mM NaCl, 2 mM DTT and incubated at RT for 15 min. The samples were, then, adsorbed to carbon-coated grids (Electron Microscopy Sciences) and stained with 0.75% uranyl formate as described[60]. Images were collected using the JEM-1400 transmission electron microscope (JEOL) at a magnification of 30,000x.

## GST pulldown
GST-tagged PKR (K296R) (0.7 μM) and PACT (5 μM) were individually benzonase-treated for 15 min at RT in the presence of 1 mM MgCl$_2$. Following the benzonase treatment, both proteins were mixed together and incubated for another 15 min at RT. The protein mix was then incubated at 37 °C for 10 min in 20 mM HEPES pH 7.5, 150 mM NaCl, 2 mM DTT before adding to Glutathione Sepharose 4B beads (Cytiva) pre-equilibrated with 50 mM Tris pH 7.5, 250 mM NaCl and incubated at 4 °C overnight with gentle rocking. The beads were then washed

3 times with 50 mM Tris pH 7.5, 250 mM NaCl before eluting with 50 mM reduced glutathione in 50 mM Tris, 150 mM NaCl. The samples were subjected to SDS-PAGE and analyzed by Western blot. The method for PKR pulldown by PACT D1D2-GST was the same as above except that 0.45 mg/ml PACT D1D2-GST and 0.35 mg/ml of PKR (K296R), full-length or ΔDRBD, were used.

## Microscale Thermophoresis

PACT ΔD3 was purified using the method described earlier, but a fluorescent label was added prior to the heparin chromatography step. This was achieved by incubating the protein with 1 mg/ml *S. aureus* sortase A (a gift from H. Ploegh, MIT)[61] and 100 μM peptide (LPETGG) conjugated with FAM (Anaspec) for 2 h at RT protected from light. The FAM-labeled PACT ΔD3 GST-tagged PKR (K296R) was pre-treated with benzonase at RT for 15 min in the presence of 1 mM $MgCl_2$ to remove any RNA contaminants. Post-benzonase treatment, FAM-labeled PACT ΔD3 (1 μM) was incubated with 16 different concentrations of GST-tagged PKR (K296R) (5 μM to 0.000458 μM) in 20 mM HEPES pH 7.5, 150 mM NaCl, 2 mM DTT, 0.05% Tween20 for 30 min at RT in the dark. Each titration mix was loaded in a separate capillary, and the MST scan was recorded on a Monolith NT.115pico (NanoTemper Technologies) using the nano Blue detector at the Center for Macromolecular Interactions, Harvard Medical School. Raw fluorescence for each capillary was scanned before and after the MST trace to ensure that the samples did not aggregate during the course of the MST experiment. The data from 3 independent experiments were analyzed using MO. Affinity Analysis v2.3.

## Bioanalyzer analysis

Total RNA from 5 different cell lines (HCC1806, NCI-H727, NCI-H2286, NCI-H2023, NCI-H1437) was isolated on day 6 after control or PACT knockout. For RNA isolation, TRIzol reagent (Thermo) was added to the cells after removing media and washing once with phosphate buffer saline (PBS). The TRIzol-mixed cell lysate was used for RNA extraction using Directzol RNA miniprep kit (Zymo Research). The purified RNA was loaded on an RNA nano chip using an Agilent Bioanalyzer.

## NMR

**Sample preparation.** PACT D3 triple labeled ($^{15}$N, $^{13}$C, ~85% $^2$H) sample was prepared by expressing the protein in *E. coli* BL21(DE3) grown in M9 minimal media supplemented with $^{15}$NH$_4$Cl (99%), $^{13}$C D-glucose (99%) as the sole sources of N and C, respectively. The minimal media was made in deuterium oxide instead of $H_2O$ in order to ensure $^2$H incorporation in the protein. The purification method was the same as described in the previous section, except that final SEC was done in 20 mM HEPES pH 7.5, 50 mM KCl, 2 mM DTT. For a mixed isotope sample, 2 separate preparations were carried out: one with $^{15}$NH$_4$Cl (99%) in deuterium oxide M9 minimal media for $^{15}$N, $^2$H-labeled protomer sample, and the other with $^{13}$C D-glucose (99%) in regular M9 minimal media for $^{13}$C-labeled protomer. Both protomers were individually purified and then denatured by adding 6 volumes of 7 M guanidinium hydrochloride along with 20 mM BME and shaking at 37 °C for 1 h. The two denatured protomers were then mixed in an equimolar ratio and dialyzed overnight in 50 mM HEPES pH 7.5, 150 mM NaCl, 2 mM DTT at 4 °C using a 3 kDa MWCO. The refolded mixed isotope sample was concentrated and subjected to SEC in 20 mM HEPES, pH 7.5, 50 mM KCl, 2 mM DTT.

**Assignment of NMR resonances and NOE restraints.** All NMR data were recorded at 30 °C (303 K) on Bruker spectrometers operating at $^1$H frequency of 900 MHz or 600 MHz and equipped with cryogenic probes. NMR data were processed using NMRPipe[62], and spectra analysis was performed using XEASY[63] and CcpNmr[64]. Triple resonance experiments were collected at $^1$H frequency of 600 MHz using a ($^{15}$N,

$^{13}$C, ~85% $^2$H)-labeled sample. Sequence-specific assignment of backbone chemical shifts was accomplished using three pairs of TROSY-enhanced triple resonance experiments[65,66], including HNCA, HN(CO)CA, HN(CA)CO, HNCO, HNCACB, and HN(CO)CACB. The aliphatic and aromatic resonances of the protein sidechains were assigned using the 3D $^{15}$N-edited NOESY-TROSY-HSQC ($\tau_{NOE}$ = 80 ms) and 3D $^{13}$C-edited NOESY-HSQC ($\tau_{NOE}$ = 150 ms) spectra, recorded at $^1$H frequency of 900 MHz using a ($^{15}$N, $^{13}$C)-labeled protein sample. These NOE spectra were also used for assigning intra-chain NOEs. For assigning inter-chain distance restraints, a mixed sample was prepared in which half of the protomers were $^{15}$N, $^2$H-labeled (0.4 mM) and the other half 15% $^{13}$C-labeled (0.4 mM). Recording a 3D $^{15}$N-edited NOESY-TROSY-HSQC ($\tau_{NOE}$ = 200 ms) on this sample allowed measurement of exclusive NOEs between the $^{15}$N-attached protons of one subunit and aliphatic protons of the neighboring subunits. The non-deuterated protein was 15% $^{13}$C-labeled for recording the $^1$H-$^{13}$C HSQC spectrum as an internal aliphatic proton chemical shift reference.

**Structure calculation.** Structure calculation was performed using the program XPLOR-NIH[67], involving two steps: (1) obtaining the structure of a monomeric subunit and (2) determining the dimeric assembly that best satisfies the intermolecular NOE restraints. In the first step, we calculated the monomer structure using all intramolecular NOE restraints, hydrogen bond restraints, and backbone dihedral angles derived from $^1$H$_N$, $^{15}$N, $^{13}$C$_\alpha$, $^{13}$C$_\beta$, and $^{13}$C' chemical shifts using the TALOS+ program[68]. For this step, we used a simulated annealing (SA) protocol in which the temperature in the bath was cooled from 2000 to 200 K with steps of 40 K. The distance restraints were enforced by flat-well harmonic potentials, with the force constant ramped from 2 to 30 kcal/mol Å$^{-2}$ during annealing. Backbone dihedral angle restraints were taken from the "GOOD" dihedral angles from TALOS+, all with a flat-well (±the corresponding uncertainties from TALOS+) harmonic potential with force constant ramped from 30 to 100 kcal/mol rad$^{-2}$. A total of 50 structures were calculated, and the lowest energy structure was selected as the monomer structure for subsequent dimer structure calculation. For dimer calculation, we generated two identical copies of the monomer structure from Step 1 and performed a similar SA protocol while applying all NMR restraints, including the inter-chain distance restraints. In this protocol, the bath was cooled from 1000 to 200 K with steps of 20 K. The NMR restraints were enforced as in Step 1. A total of 75 structures were calculated, and 15 lowest energy structures were selected as the final structural ensemble.

**Single-molecule FRET measurements.** For 3'-labeling with Cy3 or biotin, 112 nt RNA was oxidized with 0.1 M sodium meta-periodate (Pierce) overnight in 0.1 M NaOAc pH 5.4. The reaction was quenched with 250 mM KCl, buffer exchanged using Zaba desalting columns (Thermo Fischer) into 0.1 M NaOAc pH 5.4, and further incubated with 0.1 M Cy3-Mono-hydrazide (GE Healthcare) or EZ-Link Hydrazide-PEG4-Biotin (Pierce) for Cy3 or biotin labeling, respectively, for 4–6 h at RT. The dsRNA sample was then prepared by annealing 3'-Cy3 forward strand (112nt) and 3'-biotin labeled reverse complementary strand by mixing them in equimolar ratio to 95 °C followed by slowly cooling down to 25 °C at a ramp speed of −0.5 °C/s. PKR K296R mutant or PACT was labeled with Cy5 using Sortase A as described above.

All the single-molecule experiments were carried out using a home-built prism-type total internal reflection fluorescence microscope at room temperature (23.0 ± 1.0 °C). Cy3 labeled dsRNA was diluted to 200 pM with 20 μL of 10 mM pH 8.5 Tris-HCl and 50 mM NaCl (T50) and immobilized on a PEG-coated quartz slide pretreated with neutravidin (0.05 mg/ml). To minimize bleaching and blinking of dyes, an imaging buffer was prepared with reaction buffer (1 mg/ml glucose oxidase, 0.8% v/v dextrose, ~2 mM Trolox, 0.03 mg/ml catalase, 20 mM HEPES pH 7.5, and 50 mM NaCl). Five-hundred thirty-two-nanometer solid-state diode laser was employed to excite Cy3.

Fluorescent emissions of Cy3 and Cy5 were separated by a dichroic mirror (cutoff: 640 nm) and collected by electron-multiplying CCD (Andor). All data were recorded with a 33 ms time resolution by smCamera software and analyzed with Interactive Data Language. Single-molecule traces and histograms are further analyzed with scripts written in MATLAB. IDL and MATLAB scripts were provided by Dr.Taekjip Ha's lab. The duration time and sliding time were fitted using Origin 2018.

**Data analysis for single-molecule FRET.** The auto-correlation function was used to analyze the sliding time of PKR, PACT, and PKR in the presence of PACT, as described in ref. 54. The equation of the auto-correlation function is as below:

$$G(\tau) = \int E(\tau) \cdot E(t-\tau)dt \qquad (2)$$

$E(\tau)$ represents FRET, and the FRET auto-correlation, $G(\tau)$, gave us an average sliding time because FRET changes in our system reflect the diffusion of a protein. More than 200 binding events with 3–5 movies were analyzed for each autocorrelation curve, and it was fitted with a single exponential decay to obtain an average sliding time of the protein. Duration times of the diffusive state were measured from individual traces and were plotted by Prism10. To compare the movement range of PKR with and without PACT, we performed the histogram analysis of FRET states. Aggregate analysis of all traces was unable to distinguish between movements confined to low-, mid-, or high-FRET state, vs. those that fluctuate between low and high states. Therefore, we limited our analysis to traces showing high FRET and compiled these traces to generate the FRET histogram for 50 nM PKR with and without PACT.

**Analysis of sequencing data**

**RNA-seq alignment and normalization.** Pair-ended fastq samples were first trimmed by trimmomatic (v0.36)[69], and mapped to *Homo sapiens* RNA45SN1 (NCBI Gene ID: 106631777) and RNA5S1 (NCBI Gene ID: 100169751). The leftover reads were then mapped to the human genome (GRCh38.p14). Only primary alignments were kept by SAMtools (v1.9)[70] flag -F 260. To adjust for different sequencing depths among samples, the bam files were first converted to bedgraphs by bedtools genomecov (v2.30.0)[71], and the bedgraph coverage values were normalized to "as if one million total non-rRNA reads were mapped to human genome" by multiplying the coverage value with a factor of $10^6$/(total non-rRNA mapped read counts). The normalized bedgraphs were named as "*PM.bedgraph" for individual groups in our GEO submission.

**Bedgraph peak calling and AuC calculation.** The normalized fCLIP bedgraphs were first subtracted from normalized input bedgraphs with macs2 (v2.1.1.20160309) bdgcmp[72], and the resultant bedgraphs were used for peak calling with macs2 bdgpeakcall[72]. The individual AuCs were calculated from these peak regions using bedtools (v2.30.0) intersect with individual PM bedgraphs, and then calculating the sum of all overlapping bedgraph regions in the same peak.

**Inverted repeat calculation**
Repeat regions in the human genome were acquired from Repbase (girinst.org/repbase). For all repeat regions of the same family, their sequences were searched for similarity by NCBI BLAST (blastn v2.15.0; reward = 2, penalty = −3, gapopen = 5, gapextend = 2, word_size = 11) with neighboring repeats up to 1.5 kb. If their whole sequences match over 80% identity with a plus/minus match, they were then identified as inverted repeats.

**Deseq2 calculation of AuC values**
The adjusted p-value and FC of AuC values in peak regions were calculated by Deseq2[73] in R. Using peak regions as the index, the Deseq2 result datasheet was merged with other associated information (e.g., inverted repeats) using peak name as index by Python pandas.

**Reporting summary**
Further information on research design is available in the Nature Portfolio Reporting Summary linked to this article.

## Data availability
The NMR data in this study is deposited Biological Magnetic Resonance Bank with the BMRB accession code: 36671. The atomic coordinates have been deposited in the Protein Data Bank with accession code 8ZU6. The raw sequencing data for the fCLIP-seq data is available at NCBI's Gene Expression Omnibus and are accessible through the GEO Series accession number GSE269684. Source data are provided with this paper.

## Code availability
Code including detailed documentation of the sbatch shell script to trim and map the fastq files using Harvard Medical School (HMS) O2 cloud computing platform, the sbatch shell script to normalize bam files, convert to bedgraph, subtract input from fCLIP, call peak and individual AuC calculation and the python script of calculating Inverted Repeat regions using NCBI BLAST engine has been deposited in the Github repository at https://github.com/DylannnWX/Hurlab_PKR_manuscript/tree/main[74].

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

## Acknowledgements

Bioanalyzer and Tape station analyzes for fCLIP-seq library quality control were done at Molecular Genetics Core Facility at Boston Children's Hospital. MST data was collected and analyzed at Center for Macro-molecular Interactions, Harvard Medical School. This work was supported by NIH grants: T32 CA009172 (T.Z.), R35 CA197568 (M.M.), R01 GM149729-04 (S.M.), GM116898 (J.C.), R01AI154653 (S.H.); the DFCI Developmental Research Project Award in Lung Cancer Research (T.Z.), and the American Cancer Society Research Professorship (M.M.). S.H. is a Howard Hughes Medical Institute investigator.

## Author contributions

S.A., T.Z., M.M., and S.H. conceived the project. S.A., T.Z., J.H., S.M., J.C., M.M., and S.H. designed the study and interpreted the results. S.A., T.Z., J.H., L.Z., A.D., P.Z., A.R.F., D.C-R., and A.O. performed experiments and analyzed data. S.A., X.W., I.B., and E.Y.L. performed computational analysis. S.A. and S.H. wrote the manuscript. All authors discussed the results and commented on the manuscript.

## Competing interests

The authors declare no competing interests.
