## [Transparent Peer Review file · Nature Communications]

PACT prevents aberrant activation of PKR by endogenous dsRNA without sequestration

Corresponding Author: Dr Sun Hur

Version 0:

Reviewer comments:

Reviewer #1

(Remarks to the Author)

Ahmad et al., present a multi-disciplinary, mechanistic analysis of how PACT regulates the kinase activity of PKR induced by endogenous dsRNAs. The authors started by demonstrating the essential function of PACT in certain cell lines and its anti-correlation with PKR activity. Using fCLIP, the authors found that PACT and PKR share endogenous dsRNAs substrates (mostly Alu elements), leading to a co-occupancy model as opposed to a competitive binding model. Using PKR kinase assays, structural modeling, and biophysical analyses, the authors further dissected the mechanisms of PACT regulation of PKR. These data convincingly support a rheostat-like, co-occupancy model where PACT employs its dsRNA-binding and PKR-binding capabilities, mediated by distinct interfaces on PACT, to disperse PKR monomers along long dsRNAs and prevent their mutual approach and activation via trans-phosphorylation. Overall, this is a study of outstanding quality. It is cleverly designed, meticulously executed, rigorously controlled, logically interpreted, and clearly written and illustrated. It uncovers a completely novel yet highly feasible mechanism that explains the complex interplay between different dsRNA-binding proteins. It represents a substantive conceptual advance in understanding endogenous dsRNA-mediated innate immunity and its endogenous regulation. I recommend this work for publication in Nat Commun with great enthusiasm. The work has very few flaws or issues. I have a few minor questions and suggestions below mostly for discussions.

1. Fig. S4D. There are some unusual features in the PACT+dsRNA panel that resembles some form of assembly or clustering, contrasting the much cleaner PKR+dsRNA panel. Do the authors believe they are background particles, or these represent some partial aggregates or assemblies mediated by PACT dimerization?
2. The authors interpret their results largely in the context of a putative dsRBD-scanning model. In theory, a dsRNA length dependency does not necessarily have to invoke scanning. Longer dsRNAs simply offer more covalently linked binding sites that would increase the chance of protein engagement and co-occupancy than shorter dsRNAs. On the flip side, if all these dsRNA-binding proteins would be constantly scanning and sliding on the dsRNAs, one would expect to see PACT impacting ADAR activity or vice versa. I wonder if the authors could discuss further whether their data directly speak to the scanning model.
3. Line 266. "Comparing 112 bp dsRNA and 388 bp perfect Alu dsRNA (without structural irregularities) showed similar levels of PKR activity (Figure 3D), suggesting that PKR length sensitivity saturates around 112 bp." I suggest the authors temper this statement, as the lengths tested are still quite sparse. For example, dsRNA lengths of 80, 200, or 250 bp could be the optimal length instead of 112 bp. Also, as the sequences changed along with the lengths, there could be other factors at play, in terms of potency towards PKR activation.
4. The authors mostly assessed the effects of PACT on PKR in the context of being activated by endogenous dsRNA (mostly Alus). Since PKR's primary role is in antiviral response, I wonder if the authors could comment in discussions on how PACT may affect PKR activity and the overall host response during viral infection?

(Remarks on code availability)

Reviewer #2

(Remarks to the Author)

In their article titled "PACT prevents aberrant activation of PKR by endogenous dsRNA without sequestration," the authors convincingly demonstrate that PACT, a purported activator of PKR, is in actuality a suppressor of PKR function. PKR is a key part of the innate antiviral immune response, and its function is vital for controlling viral spread in a host. As the host genome also encodes for transcripts which can form double-stranded RNA (dsRNA), control of PKR function is paramount to have antiviral activity in the context of infection while avoiding activation by endogenous transcripts such as those containing Alu elements. In this context, understanding how PKR is regulated by PACT has the potential to broadly influence the field of dsRNA-sensing research and has implications for potential therapeutics for a wide slate of disease. Until now, PACT's function has previously been examined in a variety of conflicting literature, and the field would benefit greatly from the findings in this manuscript showing that PACT unequivocally inhibits PKR activation *in vitro* and *in vivo*. In that sense, we believe Ahmad et al. significantly advances the field and is an impactful study fit for Nature Communications. In this manuscript, utilizing a cell culture system, the authors found that loss of PACT either had little effect on cell growth or was significantly detrimental to cell proliferation/survival depending on cell line—in this latter group, the authors found that PKR was phosphorylated and active when PACT was targeted for deletion. The authors then attempt to prove, largely through biochemical techniques, that PACT: 1. Directly interfaces with PKR, 2. Inhibits PKR activation, 3. Inhibits PKR more effectively in the context of dsRNA, but without stopping PKR from binding these same RNAs, and that this inhibition is 4. Dependent on a DRBD-Kinase domain interface. While the authors make strong strides towards justifying these conclusions, the mechanism of how PACT inhibits PKR activation still remains quite ambiguous. This manuscript would benefit from several significant revisions:

Major Points:

1. The authors utilize various lengths of dsRNAs sourced from the beginning 30/50/100 or 500 bases in the MDA5 gene. They then go on to make conclusions that PKR activation is largely dependent on dsRNA length (Fig 3), as well as that PACT inhibits more effectively at different dsRNA lengths (Fig 4). There are other differences than length in these dsRNAs, however – there are sequence changes throughout (Sup Table S3), including changes in GC content. The authors should add a control where the dsRNAs used to compare length dependence are largely similar in sequence, especially in GC content. In short, the authors need to demonstrate more conclusively that it is length of dsRNA, and not another factor (e.g., dsRNA sequence, GC content), that influences PKR activation.
2. The authors utilize PKR and PACT formaldehyde cross-linking and immunoprecipitation (fCLIP) to determine which transcripts are bound by PKR and PACT respectively, and further use this data to argue that PACT does not sequester dsRNA from PKR. However, data derived from fCLIP experiments are very hard to interpret because of the following reasons. We recommend interpreting the fCLIP data with caution, and also toning down the title which states that PACT prevents PKR activation without dsRNA sequestration.
 - A downside to using formaldehyde to crosslink is that it is not specific to RNA-protein interactions, and proteins which associate with each other will also be cross-linked. It is therefore hard to tell if PACT and PKR are directly binding these pulled-down transcripts, or if they are interacting with other proteins which then bind these transcripts.
 - The authors use the data from 2E and 2F to argue that PACT does not sequester dsRNA away from PKR, but it is difficult to make that conclusion from the data shown. While the identities of PKR-bound transcripts is largely unchanged with or without PACT complementation, this does not mean that the absolute number of transcripts bound by PKR is unchanged. Indeed, even if PACT does block PKR binding to dsRNA, we might still expect PKR to show a similar constellation of bound transcript identities. Without some way of normalizing between samples (perhaps with an ERCC spike-in control?) to approximate absolute RNA count, one cannot determine whether or not PKR is binding fewer transcripts in the presence of PACT. Instead of the RNA-Seq experiment in 2E and 2F, the RT-qPCR experiment in 2G is more convincing to this reviewer that there may not be absolute changes in PKR bound dsRNA transcripts +/- PACT as the authors propose. The analysis in 2G could be expanded, and detailed information on how 2G was performed is lacking. Especially the combined use of input and normalization to 18S in figure 2G is confusing; normalization to 18S would appear to be sufficient. Could the authors clarify the exact procedure for their experiment and analysis in 2G?
 - The use of ectopically expressed PACT or PKR might distort the repertoire of bound transcripts, especially given that ectopic expression of PKR markedly exceeds endogenous PKR levels (S3A).
3. The data included in figure 6 seems to not fully support the authors' model. The authors hypothesize, based on homology with other DRBDs as well as their own NMR model of a PACT DRBD3 dimer, that PACT binds PKR through a separate surface on the DRBD, away from the RNA-interfacing region. They introduce several point mutations in residues which they predict may abrogate PKR-PACT interactions, and indeed several mutations seem to no longer inhibit PKR activity (K173E and K187E in particular seem to not be able to inhibit PKR autoactivity). It is unclear if this change in inhibition is actually occurring by interfering with some specific PKR kinase domain-PACT binding, or if it is due to some other change in PACT structure/activity. The authors should show that their mutations are actually changing interactions with the PKR kinase domain to justify (or potentially revise) their model and conclusions. The authors could show a specific change in interaction in a number of ways—for example, they can generate the PKR kinase domain alone and check for PACT-Kinase domain interactions by analytical size exclusion chromatography. They could also consider a split-GFP assay (eg, transfect PKR Kinase Domain and PACT each tagged with one part of a split-GFP, then looking for changes in association with the different PACT mutants). These association studies would resolve much of the ambiguity in interpreting the authors' findings. 6E is potentially a significant finding uncovering that certain PACT point mutations can abrogate PACT activity in cells. For the PACT complementation studies in 6E, the authors should demonstrate that PACT is actually present in these cells by western blot, as point mutations can disrupt transcript/protein stability in a cell. They should also show p-PKR and total PKR

levels in these cells to convincingly demonstrate that the change in viability is due to a PKR-PACT mechanism, rather than some unexpected effect of mutant PACT expression.

4. The 4KE PACT mutant: In figure 2, the introduction of the 4KE PACT mutant is great, but please show by Western Blot that 4KE does not affect PACT stability/expression levels in cells. Additionally, if PACT is interfacing with PKR using the non-dsRNA binding interface, what effect on PKR activation do the authors see when using their 4KE mutant PACT in in vitro assays both with (fig. 4) and without dsRNA (fig. 5D)? Is dsRNA binding-dead 4KE PACT able to inhibit PKR in vitro?

5. Could the authors transfect dsRNA into cells, and then test if dsRNA transfection enhances PKR and PACT interaction (ideally the endogenous proteins) via split-GFP assay or proximity ligation assay? If such experiments work, it would strongly support the authors proposed model for PACT.

Minor Points:

1. There are several subfigures where seeing individual data points would be beneficial, rather than bar graphs. For example, subfigures 2G, 4C, 6C, S3B, and S3E would all benefit from plotting individual data points, in a similar style to what the authors use in figure 1.
2. There are statistical errors or ambiguities in several places. For example, in subfigure 2H the authors compare individual groups, but only describe using a one-way ANOVA test – here they should be using a corrected multiple comparisons test (eg Tukey-corrected multiple comparisons). In figure 6C the authors should certainly be using a one-way ANOVA followed by a corrected multiple comparisons test, rather multiple uncorrected t-tests.
3. Very minor point -- Are PACT KO cells in figure 1 growing slowly, or dying?
4. If ISRIB inhibition of PKR activity is not complete, then it is possible that the entire phenotype seen in figure 1 is indeed dependent on the PKR-ISR axis – the authors should soften their conclusions in lines 143~144.
5. It is sometimes unclear where the authors are using K296R mutant PKR and where they are not, and the justification for using catalytically dead PKR in experiments for figure 5 appears absent from the text. Additionally, should line 178 say “K296R” rather than “K269R”?
6. The title of figure 3 is too broad – the authors claim that “PKR activity is highly dependent on dsRNA length beyond 30bp,” but they do not actually test RNA 30bp in length. A more accurate title may be along the lines of “PKR activity is enhanced by longer dsRNA”
7. The authors should comment in the Discussion why they believe their findings are not in accord with earlier studies on PACT, which saw a PKR-activating effect rather than suppressing PKR.
8. The sentence spanning lines 216-218 is an overstatement – just because PACT does not inhibit ADAR1 does not mean that PACT is not inhibiting the binding of other dsRNAs to other RNA binding proteins.
9. The speculation in lines 245-253 seems strained – while a scanning mechanism may explain the results there is no evidence here or in the literature for PKR having scanning activity. There are other alternative mechanisms such as the sequence difference between the different length RNAs. If the authors wish to argue for dsRNA scanning by PKR they should directly demonstrate this function experimentally.
10. The interpretation of the data in lines 299-305 seems to dismiss the effect even a minor binding of PACT to dsRNA could conceivably have on PKR activation. If PACT is disrupting multimerization of PKR, then even a small amount of competition could impair PKR activation, even without a direct PKR-PACT interaction.
11. It is not clear why the experiment in figure 4D (IR-Alu) supports the conclusion that “these findings suggest that PACT’s mode of inhibition involves more than simple dsRNA competition.” Is PACT expected to bind similarly or differently on 112bp dsRNA vs. IR-Alu psmb2?
12. Should subfigure S4A say “PKR-dsRNA complexes” on the side rather than “PACT-dsRNA complexes”?
13. In lines 269/270, The authors state that IR-alus “were less potent in stimulating PKR than 112 bp dsRNA but were still more effective than 62 bp dsRNA (Figure 3D),” but figure 3D does not show a 62bp group.
14. The use of the term “rheostat” such as in line 41 seems inappropriate, as PACT seems to be functioning as a threshold-setter rather than a variable resistor.

(Remarks on code availability)

Reviewer #3

(Remarks to the Author)

In this manuscript, Ahmad et al. have reported that PACT inhibited PKR activation in both in vitro assays and various cancer cell lines. The authors have provided evidence that PACT is a direct negative regulator of PKR through a dsRNA length- and concentration-dependent protein-protein interaction mechanism.

Uncovering the potential interactions and regulatory mechanism among different innate immune sensors detecting dsRNAs is of great interest, but at this stage, this manuscript is too preliminary and conclusions are not well supported by sufficient evidence. Additionally, previous genetic studies have produced conflicting results regarding PACT’s effects on PKR activation (Ref. #33-35; Patel et al., 2000 PMID: 10988289; Ito et al., 1999 PMID: 10336432), however the mechanism for this discrepancy remains unclear.

In brief, the data and conclusions presented in the current study are not convincing to me due to over-reliance on results of overexpressed proteins, scarce of endogenous and in vitro data at single molecular level, lack of critical controls, and over

stated conclusions. Therefore, this manuscript is not a sufficient advance to warrant publication in Nature Communications. Below please see specific concerns.

Specific concerns:

1. Figure 1D: Pitfalls related to the results of PACT depletion leading to PKR activation in PACT KO-sensitive cells should be clarified. Given that PKR is one of the most important innate immune receptors that detects viral dsRNAs and response to other stresses, thus clarification of the examined effects of PACT on PKR in cells only under normal conditions is important. Moreover, although the phosphorylated PKR is increased after PACT knockout, why the expression level of PKR is decreased?
2. Figure 2A: What are the sequence and structural features of the 112 bp dsRNA? The reasons for the authors choosing this specific dsRNA for this experiment and most other examinations should be clarified.
3. Figures 2C-2G: The conclusion that PACT did not compete with PKR for endogenous RNA substrates is unconvincing. The authors identified that PACT and PKR shared endogenous dsRNA ligands by fCLIP. However, in the comparison of PKR fCLIP peaks with and without PACT, they overexpressed PKR while relying on endogenous levels of PACT. This raises uncertainty about whether there was sufficient PACT for effective competition and the expression level and subcellular localization of PKR should be examined. In addition, the fCLIP results of this study should be further re-verified due to findings in the current study are different from the previous report showing the association between PKR and transcripts from mitochondria, which developed the fCLIP assay (Kim et al., 2018 PMID: 30174290).
4. Figures 3A-3C: The diffusion model proposed by the authors based on one PKR activity assay is overly speculative and lacks direct supporting evidence. This hypothesis should be tested by additional approaches, such as measuring the koff of PKR on dsRNAs of varying lengths or placing a roadblock on dsRNA to observe if it reduces PKR survival lifetime on the substrates. Moreover, the absence of PKR oligomerization under negative stain EM does not necessarily indicate that PKR cannot multimerize along dsRNAs. It remains possible that the PKR multimers are not stable enough under these conditions. More evidence at the single molecular level should be included.
5. Figure 4E: The authors found that TRBP could inhibit PKR similarly as PACT, but not suppress RIG-I and ADAR1. Therefore they proposed that PACT inhibits PKR's activity through a mechanism more than merely blocking dsRNA binding. However, this set of data and conclusions are unconvincing due to the lack of critical controls including the purity of examined proteins, the binding status of individual proteins and stoichiometric issue between the examined proteins and substrate RNAs.
6. Figure 7B: There is no data in the manuscript to support the authors' model that the direct dsRNA binding facilitates the PACT-PKR interaction. The affinity (Figure S6A) between PACT and PKR interaction in the absence of RNA is too low to support this model; the results of PKR pull-down (Figures 5B-5D) and the PKR kinase assays (Figure 5F) with different truncated PKR proteins are also indirect evidence. Additional critical and other direct evidence should be provided.

(Remarks on code availability)

Version 1:

Reviewer comments:

Reviewer #1

(Remarks to the Author)

I believe the authors have done an outstanding job of responding to and addressing reviewer comments and concerns. I am impressed by both the quality and quantity of the new data and analyses added in revision, which have further enhanced the rigor of this study. I recommend this work for publication in Nat. Commun. in its current form.

(Remarks on code availability)

Reviewer #2

(Remarks to the Author)

Thank you for the revisions!

Could the authors clarify the following sentence "Instead, they are consistent with rapid scanning motion along the dsRNA. (lines 289~290)." Could the authors explain in more detail why the fluctuations observed in their FRET experiment suggest there is rapid scanning? How would the fluctuations look like if a protein bound RNA but there was no scanning?

(Remarks on code availability)

Reviewer #3

(Remarks to the Author)

The authors have submitted a revised manuscript that has addressed some of this reviewer's concerns and introduce an intriguing diffusion model for the interactions between PKR and PACT. However, the quality of single-molecule analysis is insufficient, and the lack of appropriate controls raises concerns about the accuracy of the data presented. Specifically:

1) In the smFRET experiment, the anti-correlated nature of the donor and acceptor signals typically suggests energy transfer. However, in Figure 3F, this anti-correlation is too weak, particularly in the representative trajectory of PACT, where a decrease in the Cy3 signal does not correspond to an increase in the Cy5 signal. This does not appear to indicate a true FRET event.

2) Unlike Ref #53 and #54, the current manuscript does not demonstrate repetitive FRET fluctuations. The expected continuous and periodic movement of the Cy3-protein along the dsRNA should produce repetitive FRET fluctuations, rather than random variations. It is unclear how the authors distinguish between background FRET fluctuations and those caused by protein diffusion. The observed fluctuations might simply result from background noise. For instance, in Figure 3F, FRET efficiency varies significantly even when PACT is not bound. To strengthen their findings, the authors should include TRBP or other dsRNA-binding proteins to confirm that they are observing mobile binding rather than static interactions.

3) Substrates with various length should be tested to confirm the diffusion of PKR and PACT, as demonstrated in Ref #53 and #54.

Overall, these concerns lead this reviewer to question the validity of the new data and make it difficult to support the conclusion that PKR and PACT are sliding along the dsRNA, which appears to be a central conclusion of this revised manuscript.

(Remarks on code availability)

Reviewer #4

(Remarks to the Author)

In this work, the authors analyze how proteins involved in the innate immune response recognize and interact with RNA. This is a very complex paper and the focus of my review is going to be on the single molecule fluorescence aspects of the work.

The central issue is that the authors use single molecule assays (FRET) to provide evidence for a scanning mechanism (this is brought up multiple times in the abstract alone). Reviewer 3 shared concerns about this data and all reviewers shared concerns over how strongly the presented data support scanning. I agree with the reviewer assessments. As presented, the single molecule data is potentially consistent with scanning but does not exclude other models. As presented, it does not strongly support the authors conclusions and is not yet suitable for publication in Nature Communications.

The central flaw in the presentation of the single molecule data is that it is presented as scanning being the de facto mechanism. Instead, the single molecule data should be used to test a scanning model. As such many of the control experiments previously used by the Myong lab to justify scanning (and referenced by Rev 3) are missing in this manuscript. These include analysis using orthogonal methods (PIFE) and determining whether or not the sliding time is dependent on RNA length. These are particularly important since the FRET traces presented in the manuscript are much noisier than those seen in Ref. 54, for example.

In addition, there are several aspects of the single molecule data and analysis that should be revisited. First, the raw microscopy data, and ideally the fluorescence trajectories, need to be uploaded to a central repository so that the analysis could be reproduced by others. Second, virtually no information is included in the manuscript on how FRET peaks were identified. This is particularly relevant for Figure 3F. In the top trace, there is a spike in fluorescence at about 8s, why is this not an on event? In the bottom trace, there are numerous oscillations in FRET starting at about 8s and lasting until the described on event. Why are these not peaks?

In addition, the correlation analysis is potentially problematic given the noisiness of the data in Figure 3F. The correlation times the authors report are quite short (0.2 s) and this seems to be on a similar time scale as the oscillation of the background noise in Figure 3F with PACT. What would result of a correlation analysis of the FRET signals/noise of the "off times" when PACT or other proteins are not bound? In addition, to reiterate the point above, information is needed in the manuscript about how peaks were identified since evidently the correlation analysis only included "200 binding events".

Finally, there are some book keeping items. First, the actual number of events analyzed in the correlation traces needs to be included (just saying over 200 is insufficient). Second, the authors should include the number of events observed and the

number of AOs/RNAs immobilized in each experiment. This is concerning a bit since I suspect the author's FOV is large with several hundred molecules per FOV. If only 200 events or so are analyzed, this suggests that binding may be somewhat rare. Third, I am highly suspect of single molecule experiments when only a handful of traces are shown. The authors should randomly select 10-20 traces from each analyzed experimental set and include them in supplemental so that readers have a better sense of the data quality. This is particularly relevant for the data in 3F with PACT which looks very noisy. It is a bit worrisome that the example trace may be the best "looking" trace they have.

(Remarks on code availability)

Version 2:

Reviewer comments:

Reviewer #3

(Remarks to the Author)

I'd like to begin by emphasizing that a large volume of data does not necessarily make a conclusion more convincing. In fact, some of the results in the previous version, such as Figure 3F, raised more questions about the reliability of the experiments and analyses (also brought up by reviewer #4 multiple times and has since been removed). In the current manuscript, the anti-correlation between the donor and acceptor is more visible. The representative smFRET time course traces support the authors' conclusion that interactions between PKR and the 112 bp dsRNA are dynamic. The additional analysis and PIFE results further strengthen these findings.

However, some of the newly included data do not seem to align with the idea of PKR sliding. In Figure S5E, the FRET efficiency remained high and stable during the "On" state for PKR binding to the 59 bp and 42 bp dsRNA, similar to the behavior of Dicer and ADAR1/2 statically binding to dsRNA (as referenced by the authors, PMID: 23251028 and 26184879). To my understanding, given the contour lengths of the 59 bp and 42 bp dsRNAs (15 nm and 11 nm, respectively), a protein footprint of 15 bp (2 nm) diffusing along the entire length of the substrate should produce a large amplitude of FRET change. In fact, when TRBP diffused on the 38 bp and 19 bp dsRNA, rapid and more obvious FRET fluctuations were observed (again, as referenced by the authors, PMID: 23251028).

If the authors insist that the time course traces in Figure S5E represent FRET fluctuations (which, to me, is not immediately apparent), they should discuss why the patterns differ so significantly between the two studies. For example, PKR cannot diffuse along the entire length of the dsRNA? If the authors do not interpret these as FRET fluctuations, they should also discuss why PKR seems to diffuse on the 112 bp dsRNA but not on the 59 bp or 42 bp dsRNAs.

These observations on 59 bp or 42 bp dsRNAs raise additional concerns about the experimental design and the interpretations of the single-molecule data. For instance, multiple PKR molecules could potentially bind to the 112 bp substrates, but the authors might only be detecting the one closest to the RNA end (with PKR further away from the end falling outside the FRET measurement range). This is plausible because the PKR concentration used in the single-molecule experiment was 50 nM, while 100 nM PKR efficiently formed dimers on 112 bp dsRNA (Figures 3 and 4). Given that PKR was labeled in the C-terminal kinase domain, which is linked to the RNA-binding domains by a large flexible linker, any conformational changes in the kinase domain—such as dimerization or dimer-dimer interactions—could potentially affect the FRET efficiency. I recognize that these concerns may not be easily addressed without further studies. But as noted by Reviewer #4, these data do not rule out other possibility and should at least be discussed in the manuscript. After all, strong claims require strong evidence.

(Remarks on code availability)

Reviewer #4

(Remarks to the Author)

I think the authors have greatly improved the manuscript and have added much appreciated additional, single molecule evidence to support their proposed mechanism.

I think the manuscript is now suitable for publication. It will be of high interest to those in many fields including RNA biology, immunity, single molecule biophysics, and protein/nucleic acid interactions.

(Remarks on code availability)

We thank all three reviewers for the helpful comments. We performed additional experiments and analyses, including single-molecule studies, to strengthen our conclusion. We also made an extensive revision in the text (colored blue) to more accurately describe our results, both original and new.

Here is the summary of new/revised figures.

Total number of new figure panels = 16

Total number of revised figure panels = 5

Figure 2B: Western blot added as suggested by Reviewer#2, comment#4.

Figure 2G: fCLIP-RTqPCR with endogenous PKR pulldown. In response to Reviewer#2, comment#2; Reviewer#3, comment#3.

Figure 2H: Repeated statistical analysis as recommended by Reviewer#2, minor comment#2.

Figure 3A: In vitro kinase assay using 4 different 112 bp dsRNA sequences with GC% ranging from 43-58%. In response to Reviewer#2, comment#1; Reviewer#3, comment#2.

Figures 3F,3G: Single-molecule FRET analysis showing dsRNA scanning activities of PKR and PACT. In response to Reviewer#1, comment#2; Reviewer#3, comment#4.

Figures 4D-G: Single-molecule FRET analysis showing dsRNA scanning by PKR in the presence and absence of PACT. In response to Reviewer#1, comment#2; Reviewer#3, comment#6.

Figure 5A: SDS-PAGE showing purity of the protein samples in response to Reviewer#3, comment#5.

Figure 6C: Plot changes to show all data points and repeated the statistical analysis as suggested by Reviewer#2, minor comments#1, 2.

Figures S3B, E, H: Plot changes to show all data points and repeated the statistical analysis as suggested by Reviewer#2, minor comments#1.

Figure S3G: Comparison of PKR fCLIP with and without PACT using 17 different house-keeping gene to rule out any normalization artefacts. In response to Reviewer#2, comments#2.

Figure S5B: In vitro kinase assay using 4 different 112 bp dsRNA sequences with GC% ranging from 43-58% in the presence and absence of PACT. In response to Reviewer#2, comment#1; Reviewer#3, comment#2.

Figure S6E, F: In vitro kinase assay (RNA-dependent and RNA-independent) with PACT WT and 4KE mutant, as suggested by Reviewer#2, comments#4.

Figure S7A, B: Western blots added as suggested by Reviewer#2, comment#3.

REVIEWER COMMENTS

Reviewer #1 (Remarks to the Author):

Ahmad et al., present a multi-disciplinary, mechanistic analysis of how PACT regulates the kinase activity of PKR induced by endogenous dsRNAs. The authors started by demonstrating the essential function of PACT in certain cell lines and its anti-correlation with PKR activity. Using fCLIP, the authors found that PACT and PKR share endogenous dsRNAs substrates (mostly Alu elements), leading to a co-occupancy model as opposed to a competitive binding model. Using PKR kinase assays, structural modeling, and biophysical analyses, the authors further dissected the mechanisms of PACT regulation of PKR. These data convincingly support a rheostat-like, co-occupancy model where PACT employs its dsRNA-binding and PKR-binding capabilities, mediated by distinct interfaces on PACT, to disperse PKR monomers along long dsRNAs and prevent their mutual approach and activation via trans-phosphorylation. Overall, this is a study of outstanding quality. It is cleverly designed, meticulously executed, rigorously controlled, logically interpreted, and clearly written and illustrated. It uncovers a completely novel yet highly feasible mechanism that explains the complex interplay between different dsRNA-binding proteins. It represents a substantive conceptual advance in understanding endogenous dsRNA-mediated innate immunity and its endogenous regulation. I recommend this work for publication in Nat Commun with great enthusiasm. The work has very few flaws or issues. I have a few minor questions and suggestions below mostly for discussions.

We thank the reviewer for the positive and thoughtful comments.

1. Fig. S4D. There are some unusual features in the PACT+dsRNA panel that resembles some form of assembly or clustering, contrasting the much cleaner PKR+dsRNA panel. Do the authors believe they are background particles, or these represent some partial aggregates or assemblies mediated by PACT dimerization?

> If the reviewer is referring to the white patches, these are due to uneven staining and do not reflect the nature of the particles. Most PACT particles are shown as small dots, consistent with distributive RNA binding as seen by EMSA (Fig 2A). Other species represent background noise as they occur rarely and not reproducibly.

2. The authors interpret their results largely in the context of a putative dsRBD-scanning model. In theory, a dsRNA length dependency does not necessarily have to invoke scanning. Longer dsRNAs simply offer more covalently linked binding sites that would increase the chance of protein engagement and co-occupancy than shorter dsRNAs. On the flip side, if all these dsRNA-binding proteins would be constantly scanning and sliding on the dsRNAs, one would expect to see PACT impacting ADAR activity or vice versa. I wonder if the authors could discuss further whether their data directly speak to the scanning model.

> We have now provided single-molecule analyses demonstrating that PKR scans along dsRNA and that PACT interferes with this scanning process (new Figure 3F-3G, 4D-4G).

Our data also indicate that simple dsRNA binding does not fully explain PACT's inhibition of PKR. Instead, a direct, albeit weak, interaction between PKR and PACT is involved. Supporting evidence includes co-purification of PKR and PACT without dsRNA (Figure 5C), inhibition of PKR by PACT without dsRNA at higher protein concentrations (Figure 5E) and phosphorylation of PACT by PKR—a process that requires direct interaction (Figures 5E, 5F, 6C). Notably, PACT phosphorylation is inversely correlated with PKR auto-phosphorylation, which contradicts what would be expected if PKR phosphorylated PACT non-specifically. If one considers PACT phosphorylation by PKR as an indicator of their interaction, this inverse relationship further suggests that this interaction is mechanistically linked to PACT's inhibition of PKR. Finally, PACT mutations in the putative interface—located away from the dsRNA-binding domains—impacted PACT's function without affecting dsRNA binding (Figure 6), supporting our model of direct protein-protein interaction for PKR inhibition. This explains why loss of PACT does not globally activate other dsRNA-binding proteins, such as ADAR1 (Figure 2H), OASes (Figure S1G), or RIG-I-like receptors (Figure S1A). Moreover, RIG-I and ADAR1 DRBDs did not inhibit PKR in vitro despite displaying similar dsRNA affinity as PACT (Figure 5A). Collectively, our results support a model in which PACT inhibits PKR by blocking its scanning activity through dsRNA-facilitated protein-protein interactions. However, we acknowledge potential alternative interpretations of our data and have provided a more detailed discussion in the revised Discussion section (line 476-488).

3. Line 266. “Comparing 112 bp dsRNA and 388 bp perfect Alu dsRNA (without structural irregularities) showed similar levels of PKR activity (Figure 3D), suggesting that PKR length sensitivity saturates around 112 bp.” I suggest the authors temper this statement, as the lengths tested are still quite sparse. For example, dsRNA lengths of 80, 200, or 250 bp could be the optimal length instead of 112 bp. Also, as the sequences changed along with the lengths, there could be other factors at play, in terms of potency towards PKR activation.

> We agree with the reviewer and have modified our conclusion to “suggesting that PKR length sensitivity saturates around 100-400 bp.” (line 258). To address the potential impact of the dsRNA sequence, we examined PKR activity with 4 different 112 bp dsRNA with varying sequences and GC contents (new Figure 3A, see Table S3 for sequences). All showed similar activities at three different dsRNA concentrations. PACT inhibition was also similarly observed across these sequences (new Figure S5B). These results suggest that both PKR and PACT have little dependence on dsRNA sequence, which is consistent with the known mode of DRBD–dsRNA interaction¹.

4. The authors mostly assessed the effects of PACT on PKR in the context of being activated by endogenous dsRNA (mostly Alus). Since PKR's primary role is in antiviral response, I wonder if the authors could comment in discussions on how PACT may affect PKR activity and the overall host response during viral infection?

> Our results suggest that PACT's activity is highly dependent on dsRNA concentration and length (Figure 4). Based on this, we expect that PACT's effect on viral infection may also be

virus- and dose-dependent, as different viruses accumulate dsRNAs of varying lengths to different extents. This is now discussed in Discussion (line 470-472).

Reviewer #2 (Remarks to the Author):

In their article titled “PACT prevents aberrant activation of PKR by endogenous dsRNA without sequestration,” the authors convincingly demonstrate that PACT, a purported activator of PKR, is in actuality a suppressor of PKR function. PKR is a key part of the innate antiviral immune response, and its function is vital for controlling viral spread in a host. As the host genome also encodes for transcripts which can form double-stranded RNA (dsRNA), control of PKR function is paramount to have antiviral activity in the context of infection while avoiding activation by endogenous transcripts such as those containing Alu elements. In this context, understanding how PKR is regulated by PACT has the potential to broadly influence the field of dsRNA-sensing research and has implications for potential therapeutics for a wide slate of disease. Until now, PACT’s function has previously been examined in a variety of conflicting literature, and the field would benefit greatly from the findings in this manuscript showing that PACT unequivocally inhibits PKR activation in vitro and in vivo. In that sense, we believe Ahmad et al. significantly advances the field and is an impactful study fit for Nature Communications.

In this manuscript, utilizing a cell culture system, the authors found that loss of PACT either had little effect on cell growth or was significantly detrimental to cell proliferation/survival depending on cell line—in this latter group, the authors found that PKR was phosphorylated and active when PACT was targeted for deletion. The authors then attempt to prove, largely through biochemical techniques, that PACT: 1. Directly interfaces with PKR, 2. Inhibits PKR activation, 3. Inhibits PKR more effectively in the context of dsRNA, but without stopping PKR from binding these same RNAs, and that this inhibition is 4. Dependent on a DRBD-Kinase domain interface. While the authors make strong strides towards justifying these conclusions, the mechanism of how PACT inhibits PKR activation still remains quite ambiguous. This manuscript would benefit from several significant revisions:

We thank the reviewer for the insightful comments and suggestions.

Major Points:

1. The authors utilize various lengths of dsRNAs sourced from the beginning 30/50/100 or 500 bases in the MDA5 gene. They then go on to make conclusions that PKR activation is largely dependent on dsRNA length (Fig 3), as well as that PACT inhibits more effectively at different dsRNA lengths (Fig 4). There are other differences than length in these dsRNAs, however – there are sequence changes throughout (Sup Table S3), including changes in GC content. The authors should add a control where the dsRNAs used to compare length dependence are largely similar in sequence, especially in GC content. In short, the authors need to demonstrate more conclusively that it is length of dsRNA, and not another factor (e.g., dsRNA sequence, GC content), that influences PKR activation.

> We agree with the reviewer and performed additional experiments to examine the potential sequence dependence of PKR and PACT. We used 4 different dsRNA of the same length (112 bp) but with varying sequences and GC contents (new Figure 3A, Figure S5B, see Table S3 for sequences). At three different dsRNA concentrations, we observed that all four dsRNAs

stimulated PKR to similar extents (Figure S5C). PACT inhibition was also similarly observed across these sequences (Figure S5C). These results suggest that both PKR and PACT have little dependence on dsRNA sequence, which is consistent with the known mode of DRBD–dsRNA interaction¹.

2. The authors utilize PKR and PACT formaldehyde cross-linking and immunoprecipitation (fCLIP) to determine which transcripts are bound by PKR and PACT respectively, and further use this data to argue that PACT does not sequester dsRNA from PKR. However, data derived from fCLIP experiments are very hard to interpret because of the following reasons. We recommend interpreting the fCLIP data with caution, and also toning down the title which states that PACT prevents PKR activation without dsRNA sequestration.

- A downside to using formaldehyde to crosslink is that it is not specific to RNA-protein interactions, and proteins which associate with each other will also be cross-linked. It is therefore hard to tell if PACT and PKR are directly binding these pulled-down transcripts, or if they are interacting with other proteins which then bind these transcripts.

>We acknowledge that fCLIP can be less specific than direct crosslinking. However, direct crosslinking is generally inefficient for dsRNA-binding proteins because their interactions are largely limited to RNA phosphate backbones, rather than chemically reactive bases. Consequently, fCLIP is commonly used for dsRNA-binding proteins, such as PKR^{2,3}, Staufen^{4,5}, Drosha⁶⁻⁸, and MDA5⁹.

To distinguish specific PKR-bound peaks from background, we compared fCLIP data from PKR-sufficient and PKR-deficient cells (see Figure 2C legend, also see Figure S3C). After filtering out peaks that are also enriched in PKR-deficient cells, most remaining peaks corresponded to IR-Alus, a known major source of endogenous dsRNA¹. Similarly, we used fCLIP from PACT-sufficient and -deficient cells, to identify IR-Alus as major ligands. The direct nature of their interactions is further supported by our biochemical and biophysical analyses using purified PKR, PACT and IR-Alus as well as other dsRNAs (Figure 3-4, Figure S4-S5). Our biochemical reconstitution and new single molecule data (Figure 3-4) also recapitulated the fCLIP finding that PACT can inhibit PKR without blocking PKR's access to dsRNA. Given the extensive in vitro characterization of PACT and PKR using purified proteins and RNA, we believe our results collectively provide sufficient support that both PKR and PACT directly interact with IR-Alus for their biological functions.

Regarding the title, we respectfully maintain that our data provide sufficient support. As mentioned above, supporting evidence is not limited to fCLIP, but also includes extensive in vitro studies assessing PKR activity across various dsRNA lengths, concentrations, and PACT levels, along with newly conducted single-molecule analyses. Specifically, single-molecule studies revealed that PACT impairs PKR's ability to scan along dsRNA without blocking RNA binding. This mechanism explains how a substoichiometric amount of PACT can inhibit PKR functions and why PKR fCLIP remains largely unaffected by the presence of PACT.

- The authors use the data from 2E and 2F to argue that PACT does not sequester dsRNA away from PKR, but it is difficult to make that conclusion from the data shown. While the identities of PKR-bound transcripts is largely unchanged with or without PACT complementation, this does not mean that the absolute number of transcripts bound by PKR is unchanged. Indeed, even if PACT does block PKR binding to dsRNA, we might still expect PKR to show a similar constellation of bound transcript identities. Without some way of normalizing between samples (perhaps with an ERCC spike-in control?) to approximate absolute RNA count, one cannot determine whether or not PKR is binding fewer transcripts in the presence of PACT. Instead of the RNA-Seq experiment in 2E and 2F, the RT-qPCR experiment in 2G is more convincing to this reviewer that there may not be absolute changes in PKR bound dsRNA transcripts +/- PACT as the authors propose. The analysis in 2G could be expanded, and detailed information on how 2G was performed is lacking. Especially the combined use of input and normalization to 18S in figure 2G is confusing; normalization to 18S would appear to be sufficient. Could the authors clarify the exact procedure for their experiment and analysis in 2G?

> We would like to first clarify that the values in original Fig 2G (now Fig S3H, with ectopically expressed PKR) and new Fig 2G (endogenous PKR) were obtained by first normalizing indicated RNA levels to the internal control 18S rRNA, followed by calculating the ratio of fCLIP to input for the normalized RNA levels. This approach aligns with traditional ChIP/RT-qPCR analysis, where ChIPed values are normalized to input values, and is particularly important when variables like the presence or absence of PACT alter the global cell state. This is now explicitly described in the figure legend.

For fCLIP-seq data, the reviewer's concern is valid and would typically require a spike-in control if fCLIP only pulled down PKR ligands. However, sufficient background exists for fCLIP and this background RNA—those that are not enriched by fCLIP or those that are enriched in fCLIP from both PKR-sufficient and -deficient cells—allowed for effective normalization. Specifically, we normalized read counts to the total non-rRNA counts. rRNAs were excluded from normalization because they were experimentally depleted during library preparation, and residual rRNA depletion efficiency could artificially influence normalization. This strategy proved effective, as evidenced by equivalent levels of 17 distinct housekeeping genes—including ACTB, GAPDH, and various ribosomal protein genes—in both PACT-sufficient and PACT-deficient cells (Figure S3G). This means that the fCLIP/input comparisons between PACT-sufficient and PACT-deficient cells in Figure 2F would yield similar results even if any housekeeping gene were used for normalization. These normalization methods are now clearly explained in the figure legends.

- The use of ectopically expressed PACT or PKR might distort the repertoire of bound transcripts, especially given that ectopic expression of PKR markedly exceeds endogenous PKR levels (S3A).

> We agree with the reviewer's concern and repeated PKR fCLIP using endogenous PKR, in the presence and absence of endogenous PACT (in new Fig 2G). The results again showed that PKR's interactions with IR-Alus are largely unaffected by PACT, supporting our original conclusion.

3. The data included in figure 6 seems to not fully support the authors' model. The authors hypothesize, based on homology with other DRBDs as well as their own NMR model of a PACT DRBD3 dimer, that PACT binds PKR through a separate surface on the DRBD, away from the RNA-interfacing region. They introduce several point mutations in residues which they predict may abrogate PKR-PACT interactions, and indeed several mutations seem to no longer inhibit PKR activity (K173E and K187E in particular seem to not be able to inhibit PKR autoactivity). It is unclear if this change in inhibition is actually occurring by interfering with some specific PKR kinase domain-PACT binding, or if it is due to some other change in PACT structure/activity. The authors should show that their mutations are actually changing interactions with the PKR kinase domain to justify (or potentially revise) their model and conclusions. The authors could show a specific change in interaction in a number of ways—for example, they can generate the PKR kinase domain alone and check for PACT-Kinase domain interactions by analytical size exclusion chromatography. They could also consider a split-GFP assay (eg, transfect PKR Kinase Domain and PACT each tagged with one part of a split-GFP, then looking for changes in association with the different PACT mutants). These association studies would resolve much of the ambiguity in interpreting the authors' findings.

6E is potentially a significant finding uncovering that certain PACT point mutations can abrogate PACT activity in cells. For the PACT complementation studies in 6E, the authors should demonstrate that PACT is actually present in these cells by western blot, as point mutations can disrupt transcript/protein stability in a cell. They should also show p-PKR and total PKR levels in these cells to convincingly demonstrate that the change in viability is due to a PKR-PACT mechanism, rather than some unexpected effect of mutant PACT expression.

>We now provide the WB data showing the level of PACT (WT and mutants) as well as p-PKR in our complementation analyses using NCI-H727 and HCC1806 cells (new Figure S7). The results clearly demonstrate that the K187E and K191E mutants do not exhibit compromised stability. In fact, even when K187E and K191E are expressed at higher levels than WT PACT, they are less effective at inhibiting PKR. Additionally, we have shown that these mutants bind dsRNA as robustly as WT PACT (Figure 6B), indicating that the mutations impair PACT's function in PKR inhibition without affecting protein stability or dsRNA binding.

Regarding the interaction between PKR and PACT, quantitatively comparing WT PACT to mutants proved challenging due to their weak interactions, as noted in the original manuscript. We also attempted the recommended split GFP analysis, but the results were inconclusive, likely due to the difficulty of capturing transient interactions. However, our biochemical analyses indeed suggest that the mutants are less efficient at interacting with PKR. Figures 6C–6D showed that PACT is phosphorylated by PKR—a process requiring direct interaction—and that PACT phosphorylation inversely correlates with the level of PKR activity (as measured by PKR auto-phosphorylation). This contradicts non-specific PKR-mediated phosphorylation of PACT and supports our model of transient PKR-PACT interactions that impede PKR activation. Nonetheless, we acknowledge the limitations and revised the manuscript to recognize that alternative interpretations of our findings remain possible (lines 476-488).

4. The 4KE PACT mutant: In figure 2, the introduction of the 4KE PACT mutant is great, but please show by Western Blot that 4KE does not affect PACT stability/expression levels in cells. Additionally, if PACT is interfacing with PKR using the non-dsRNA binding interface, what effect on PKR activation do the authors see when using their 4KE mutant PACT in in vitro assays both with (fig. 4) and without dsRNA (fig. 5D)? Is dsRNA binding-dead 4KE PACT able to inhibit PKR in vitro?

> We now include the WB data showing the level of PACT (WT and 4KE) as well as p-PKR in new Figure 2B (bottom). The result clearly shows that, even when 4KE is expressed at higher levels than WT PACT, it cannot inhibit PKR. Our in vitro analyses further show that 4KE fails to inhibit PKR both in the presence and absence of dsRNA (new Figures S6E–S6F). The inability of 4KE to block dsRNA-independent PKR activity suggests that 4KE may be defective not only in dsRNA binding but also in PKR binding, as supported by the limited phosphorylation of 4KE by PKR (Figure S6F).

5. Could the authors transfect dsRNA into cells, and then test if dsRNA transfection enhances PKR and PACT interaction (ideally the endogenous proteins) via split-GFP assay or proximity ligation assay? If such experiments work, it would strongly support the authors proposed model for PACT.

> As discussed above, we attempted the recommended split GFP analysis, but the results were inconclusive. Instead, we present new single-molecule analyses (new Figures 3F-3G, 4D-4G), which demonstrate that PACT interferes with PKR's dsRNA scanning at the level of individual dsRNA molecules. Given the low affinity interaction between PACT and PKR in the absence of dsRNA, the simplest explanation for these observations is that PACT inhibits PKR's activity when both are bound to the same dsRNA.

Minor Points:

1. There are several subfigures where seeing individual data points would be beneficial, rather than bar graphs. For example, subfigures 2G, 4C, 6C, S3B, and S3E would all benefit from plotting individual data points, in a similar style to what the authors use in figure 1.

> We have modified the plots to show individual data points.

2. There are statistical errors or ambiguities in several places. For example, in subfigure 2H the authors compare individual groups, but only describe using a one-way ANOVA test – here they should be using a corrected multiple comparisons test (eg Tukey-corrected multiple comparisons). In figure 6C the authors should certainly be using a one-way ANOVA followed by a corrected multiple comparisons test, rather multiple uncorrected t-tests.

> We have modified the statistical analyses as recommended.

3. Very minor point -- Are PACT KO cells in figure 1 growing slowly, or dying?

> We believe there is a combination of cell death and growth inhibition, but we did not separately measure the two in this manuscript.

4. If ISRIB inhibition of PKR activity is not complete, then it is possible that the entire phenotype seen in figure 1 is indeed dependent on the PKR-ISR axis – the authors should soften their conclusions in lines 143~144.

>We appreciate the reviewer's feedback, but would like more guidance on how we can improve. Our data showed that PKR KO fully rescues cell viability (Fig 1E), while ISRIB partially restores it (Fig 1G). This suggests that PKR is a primary driver of cell death, but ISR downstream of PKR is only partially involved. Based on these findings, we originally stated that "PKR regulates cell fitness through both ISR-dependent and -independent mechanisms (Figures 1G, S1E-F)." If the reviewer has specific suggestions, we would be happy to consider them.

5. It is sometimes unclear where the authors are using K296R mutant PKR and where they are not, and the justification for using catalytically dead PKR in experiments for figure 5 appears absent from the text. Additionally, should line 178 say "K296R" rather than "K269R"?

>We apologize for the oversight. We corrected our error, and specified K296R mutant whenever used, and provide the rationale for why the mutant was used instead of wild-type. In most cases, PKR 296R was used to ensure analysis of PKR movement on dsRNA or interaction with PACT before its activation. We also clarified whether WT or K296R is used by stating in Figure 3 and 5 legends that "Unless mentioned otherwise, wild-type full-length PKR was used."

6. The title of figure 3 is too broad – the authors claim that "PKR activity is highly dependent on dsRNA length beyond 30bp," but they do not actually test RNA 30bp in length. A more accurate title may be along the lines of "PKR activity is enhanced by longer dsRNA"

>We now changed the subheading to "PKR scans along dsRNA, resulting in dsRNA length-dependent activity."

7. The authors should comment in the Discussion why they believe their findings are not in accord with earlier studies on PACT, which saw a PKR-activating effect rather than suppressing PKR.

>This was discussed in Results in the original manuscript, but is now also described in the Discussion section (lines 453-456).

8. The sentence spanning lines 216-218 is an overstatement – just because PACT does not inhibit ADAR1 does not mean that PACT is not inhibiting the binding of other dsRNAs to other RNA binding proteins.

>We have revised the text to clarify that we are referring to not only the ADAR1 result, but also the RLR, OAS results in Figure S1 (line 217-221).

9. The speculation in lines 245-253 seems strained – while a scanning mechanism may explain the results there is no evidence here or in the literature for PKR having scanning activity. There are other alternative mechanisms such as the sequence difference between the different length RNAs. If the authors wish to argue for dsRNA scanning by PKR they should directly demonstrate this function experimentally.

>We now provide single molecule analysis showing that PKR scans along dsRNA and PACT interferes with PKR's dsRNA scanning activity (new Figures 3F-3G, 4D-4G). We also show that neither PKR's autophosphorylation nor PACT's inhibition of PKR is dependent on dsRNA sequence by comparing 4 dsRNAs of different sequences and GC contents (Figures 3A, S5B).

10. The interpretation of the data in lines 299-305 seems to dismiss the effect even a minor binding of PACT to dsRNA could conceivably have on PKR activation. If PACT is disrupting multimerization of PKR, then even a small amount of competition could impair PKR activation, even without a direct PKR-PACT interaction.

>We would like to clarify that this paragraph is to exclude the possibility that PACT's inhibition is mediated by sequestering dsRNA away from PKR. We have revised the opening question (in line 324) to clarify this point.

11. It is not clear why the experiment in figure 4D (IR-Alu) supports the conclusion that “these findings suggest that PACT's mode of inhibition involves more than simple dsRNA competition.” Is PACT expected to bind similarly or differently on 112bp dsRNA vs. IR-Alu psmb2?

>We agree that the conclusion is not directly related to the cited results. We have removed this sentence. Given that PACT's impact on PKR is similar for both 112 bp dsRNA and IR-Alu (Figure S5C), we expect similar behavior of PACT on both dsRNAs.

12. Should subfigure S4A say “PKR-dsRNA complexes” on the side rather than “PACT-dsRNA complexes”?

>It is corrected now.

13. In lines 269/270, The authors state that IR-alus “were less potent in stimulating PKR than 112 bp dsRNA but were still more effective than 62 bp dsRNA (Figure 3D),” but figure 3D does not show a 62bp group.

>The reviewer is correct in that we cannot directly compare IR-Alus to 62 bp dsRNAs as the experiments on Fig 3C and 3D were done separately. Thus, we removed the part where we compared those two RNAs.

14. The use of the term “rheostat” such as in line 41 seems inappropriate, as PACT seems to be functioning as a threshold-setter rather than a variable resistor.

>We agree with the reviewer and replaced the term “rheostat” by “threshold-setter”.

Reviewer #3 (Remarks to the Author):

In this manuscript, Ahmad et al. have reported that PACT inhibited PKR activation in both in vitro assays and various cancer cell lines. The authors have provided evidence that PACT is a direct negative regulator of PKR through a dsRNA length- and concentration-dependent protein-protein interaction mechanism.

Uncovering the potential interactions and regulatory mechanism among different innate immune sensors detecting dsRNAs is of great interest, but at this stage, this manuscript is too preliminary and conclusions are not well supported by sufficient evidence. Additionally, previous genetic studies have produced conflicting results regarding PACT's effects on PKR activation (Ref. #33-35; Patel et al., 2000 Sua: 10988289; Ito et al., 1999 PMID: 10336432), however the mechanism for this discrepancy remains unclear.

>We would like to thank the reviewer for acknowledging that the manuscript is addressing an important issue of how dsRNA sensing pathways are regulated. However, we respectfully disagree with the reviewer that the previous genetic studies are in conflict with our results. Among the references mentioned above, Refs #33-35 are genetic studies that clearly indicate an inhibitory role of PACT on PKR via PACT-KO or KD. The other 2 references namely Patel et al., 2000 (PMID: 10988289) and Ito et al., 1999 (PMID: 10336432) do not report genetic effect of PACT KO, but rather showing that overexpressed PACT interacted with PKR under stress conditions. The reports proposing PACT as an activator of PKR have mostly been based on biochemical studies, which we believe may have resulted from incomplete removal of contaminating RNA from PACT. Our biochemical reconstitution of PKR inhibition by PACT required extensive optimization of the protein purification strategy, which is discussed in lines 306-310 and Methods. We believe our data combining bulk biochemistry, single-molecule analyses and cellular/genetic analyses, clearly demonstrate that PACT is an inhibitor of PKR.

In brief, the data and conclusions presented in the current study are not convincing to me due to over-reliance on results of overexpressed proteins, scarce of endogenous and in vitro data at single molecular level, lack of critical controls, and over stated conclusions. Therefore, this manuscript is not a sufficient advance to warrant publication in Nature Communications. Below please see specific concerns.

Specific concerns:

1. Figure 1D: Pitfalls related to the results of PACT depletion leading to PKR activation in PACT KO-sensitive cells should be clarified. Given that PKR is one of the most important innate immune receptors that detects viral dsRNAs and response to other stresses, thus clarification of the examined effects of PACT on PKR in cells only under normal conditions is important. Moreover, although the phosphorylated PKR is increased after PACT knockout, why the expression level of PKR is decreased?

> We would like to note that multiple studies have shown PKR activation by endogenous dsRNA under dysregulated states, such as loss of ADAR1¹⁰⁻¹⁴ or XRN1^{15,16}. Our study identifies PACT as

another key homeostatic regulator of PKR, demonstrating that PACT deficiency leads to aberrant PKR activation by endogenous IR-Alus. This finding adds to the growing evidence linking improper PKR activation to the pathogenesis of various diseases in the absence of viral infection. Therefore, we believe investigating the mechanisms by which PKR is regulated under normal conditions and how this regulation could get disrupted are both important and timely.

The apparent loss of total PKR upon its activation may be due to the decrease in total PKR level or excessive phosphorylation leading to evasion of PKR detection by the anti-PKR antibody. Regardless, the result clearly supports activation of PKR upon PACT depletion, as measured by both p-PKR and p-eIF2a. This increase in p-eIF2a was lost upon knocking out PKR (Figure S1B), demonstrating that PKR, not other ISR kinases, is responsible for p-eIF2a.

2. Figure 2A: What are the sequence and structural features of the 112 bp dsRNA? The reasons for the authors choosing this specific dsRNA for this experiment and most other examinations should be clarified.

> We now include new data comparing the PKR activity using 4 different 112 bp dsRNAs with varying sequences and GC contents (new Figure S5B, see Table S3 for sequences), comparing at three different RNA concentrations, with and without PACT. The result shows that both PKR and PACT are independent of dsRNA sequence, consistent with the known mode of DRBD-dsRNA interactions¹. This justifies our use of 112 bp (seq 1) for the most of our biochemical analyses.

3. Figures 2C-2G: The conclusion that PACT did not compete with PKR for endogenous RNA substrates is unconvincing. The authors identified that PACT and PKR shared endogenous dsRNA ligands by fCLIP. However, in the comparison of PKR fCLIP peaks with and without PACT, they overexpressed PKR while relying on endogenous levels of PACT. This raises uncertainty about whether there was sufficient PACT for effective competition and the expression level and subcellular localization of PKR should be examined. In addition, the fCLIP results of this study should be further re-verified due to findings in the current study are different from the previous report showing the association between PKR and transcripts from mitochondria, which developed the fCLIP assay (Kim et al., 2018 PMID: 30174290).

> We agree with the reviewer's critique on the use of ectopically expressed PKR and repeated the PKR fCLIP experiment using endogenous PKR (with and without endogenous PACT) (in new Fig 2G). The results confirmed that PACT does not impair PKR's interactions with IR-Alus, supporting our original conclusion. In addition to the fCLIP data, we present other data supporting our conclusion that PACT can inhibit PKR without sequestering dsRNA. These include our in vitro biochemical studies showing that a substoichiometric amount of PACT can inhibit PKR (Figures 4A-4C), and new single-molecule analyses showing that PACT impairs PKR's ability to scan along dsRNA without preventing its binding (Figures 4D-4G).

Regarding the discrepancy between the previous report on the importance of mitochondrial dsRNA³, we would like to clarify that this paper (PMID: 30174290), together with another paper from the same lab², in fact showed that IR-Alus are another major source of dsRNA, albeit not

as significant as mitochondrial RNA in HeLa cells. Importantly, Dorrity et al¹⁷ compared the source of dsRNA in multiple cell lines, and found that the relative contribution of mitochondrial RNA varies. Thus, we believe that our findings are in line with the published work and our biochemical studies supporting the robust PKR activation by IR-Alus.

4. Figures 3A-3C: The diffusion model proposed by the authors based on one PKR activity assay is overly speculative and lacks direct supporting evidence. This hypothesis should be tested by additional approaches, such as measuring the koff of PKR on dsRNAs of varying lengths or placing a roadblock on dsRNA to observe if it reduces PKR survival lifetime on the substrates. Moreover, the absence of PKR oligomerization under negative stain EM does not necessarily indicate that PKR cannot multimerize along dsRNAs. It remains possible that the PKR multimers are not stable enough under these conditions. More evidence at the single molecular level should be included.

>We now provide single molecule analysis showing that PKR scans along dsRNA (similar to other proteins with tandem DRBDs^{18,19}) and PACT interferes with PKR's dsRNA scanning activity (new Figures 3F-3G, 4D-4G). Regarding the multimerization statement, we agree that our data do not rule out transient multimerization on dsRNA, and thus have revised the main text to clearly state so (line 268).

5. Figure 4E: The authors found that TRBP could inhibit PKR similarly as PACT, but not suppress RIG-I and ADAR1. Therefore they proposed that PACT inhibits PKR's activity through a mechanism more than merely blocking dsRNA binding. However, this set of data and conclusions are unconvincing due to the lack of critical controls including the purity of examined proteins, the binding status of individual proteins and stoichiometric issue between the examined proteins and substrate RNAs.

>We now provide the purity of the proteins used in Fig. 5A. The dsRNA binding patterns of PACT, TRBP, RIG-I and ADAR1 were shown in the original Figure S5C, S5D (now Fig 5B, S6A). These data show that their affinities are comparable, and all of them can bind multiple sites on dsRNA.

6. Figure 7B: There is no data in the manuscript to support the authors' model that the direct dsRNA binding facilitates the PACT-PKR interaction. The affinity (Figure S6A) between PACT and PKR interaction in the absence of RNA is too low to support this model; the results of PKR pull-down (Figures 5B-5D) and the PKR kinase assays (Figure 5F) with different truncated PKR proteins are also indirect evidence. Additional critical and other direct evidence should be provided.

> Our new single-molecule analyses (new Figures 3F-3G, 4D-4G) demonstrate that PACT interferes with PKR's dsRNA scanning at the level of individual dsRNA molecules. Given the low affinity interaction between PACT and PKR in the absence of dsRNA, the simplest explanation for these observations is that PACT inhibits PKR's activity when both are bound to the same dsRNA.

If the reviewer's concern is about whether direct PKR-PACT interaction in fact occurs during PACT-mediated inhibition, we would like to summarize our supporting evidence. This includes co-purification of PKR and PACT without dsRNA (Figure 5C), inhibition of PKR by PACT without dsRNA at higher protein concentrations (Figure 5E), and phosphorylation of PACT by PKR—a process that requires direct interaction (Figures 5E, 5F, 6C). Notably, PACT phosphorylation is inversely correlated with PKR auto-phosphorylation, which contradicts what would be expected if PKR phosphorylated PACT non-specifically. If one considers PACT phosphorylation by PKR as an indicator of their interaction, this inverse relationship further suggests that this interaction is mechanistically linked to PACT's inhibition of PKR. Finally, PACT mutations in the putative interface—located away from the dsRNA-binding domains—impacted PACT's function without affecting dsRNA binding (Figure 6), supporting our model of direct protein-protein interaction for PKR inhibition. This explains why the loss of PACT does not globally activate other dsRNA-binding proteins, such as ADAR1 (Figure 2H), OASes (Figure S1G), or RIG-I-like receptors (Figure S1A). Collectively, our results are consistent with the model in which PACT inhibits PKR by blocking its scanning activity through dsRNA-facilitated protein-protein interactions. However, we acknowledge potential alternative interpretations of our data and have provided a more detailed discussion in the revised Discussion section (line 476-488).

- 1 Hur, S. Double-Stranded RNA Sensors and Modulators in Innate Immunity. *Annu Rev Immunol* **37**, 349-375 (2019). <https://doi.org/10.1146/annurev-immunol-042718-041356>
- 2 Kim, Y. *et al.* PKR is activated by cellular dsRNAs during mitosis and acts as a mitotic regulator. *Genes Dev* **28**, 1310-1322 (2014). <https://doi.org/10.1101/gad.242644.114>
- 3 Kim, Y. *et al.* PKR Senses Nuclear and Mitochondrial Signals by Interacting with Endogenous Double-Stranded RNAs. *Mol Cell* **71**, 1051-1063 e1056 (2018). <https://doi.org/10.1016/j.molcel.2018.07.029>
- 4 Ricci, E. P. *et al.* Staufen1 senses overall transcript secondary structure to regulate translation. *Nat Struct Mol Biol* **21**, 26-35 (2014). <https://doi.org/10.1038/nsmb.2739>
- 5 Elbarbary, R. A., Li, W., Tian, B. & Maquat, L. E. STAU1 binding 3' UTR IRAlus complements nuclear retention to protect cells from PKR-mediated translational shutdown. *Genes Dev* **27**, 1495-1510 (2013). <https://doi.org/10.1101/gad.220962.113>
- 6 Kim, B., Jeong, K. & Kim, V. N. Genome-wide Mapping of DROSHA Cleavage Sites on Primary MicroRNAs and Noncanonical Substrates. *Mol Cell* **66**, 258-269 e255 (2017). <https://doi.org/10.1016/j.molcel.2017.03.013>
- 7 Kim, B. & Kim, V. N. fCLIP-seq for transcriptomic footprinting of dsRNA-binding proteins: Lessons from DROSHA. *Methods* **152**, 3-11 (2019). <https://doi.org/10.1016/j.ymeth.2018.06.004>
- 8 Noguchi, S. *et al.* Cytoplasmic and nuclear DROSHA in human villous trophoblasts. *J Reprod Immunol* **162**, 104189 (2024). <https://doi.org/10.1016/j.jri.2023.104189>

- 9 Guney, M. H. *et al.* IFIH1 (MDA5) is required for innate immune detection of intron-containing RNA expressed from the HIV-1 provirus. *Proc Natl Acad Sci U S A* **121**, e2404349121 (2024). <https://doi.org/10.1073/pnas.2404349121>
- 10 Chung, H. *et al.* Human ADAR1 Prevents Endogenous RNA from Triggering Translational Shutdown. *Cell* **172**, 811-824 e814 (2018). <https://doi.org/10.1016/j.cell.2017.12.038>
- 11 Hu, S. B. *et al.* ADAR1p150 prevents MDA5 and PKR activation via distinct mechanisms to avert fatal autoinflammation. *Mol Cell* **83**, 3869-3884 e3867 (2023). <https://doi.org/10.1016/j.molcel.2023.09.018>
- 12 Gannon, H. S. *et al.* Identification of ADAR1 adenosine deaminase dependency in a subset of cancer cells. *Nat Commun* **9**, 5450 (2018). <https://doi.org/10.1038/s41467-018-07824-4>
- 13 Ishizuka, J. J. *et al.* Loss of ADAR1 in tumours overcomes resistance to immune checkpoint blockade. *Nature* **565**, 43-48 (2019). <https://doi.org/10.1038/s41586-018-0768-9>
- 14 Kung, C. P. *et al.* Evaluating the therapeutic potential of ADAR1 inhibition for triple-negative breast cancer. *Oncogene* **40**, 189-202 (2021). <https://doi.org/10.1038/s41388-020-01515-5>
- 15 Zou, T. *et al.* XRN1 deletion induces PKR-dependent cell lethality in interferon-activated cancer cells. *Cell Rep* **43**, 113783 (2024). <https://doi.org/10.1016/j.celrep.2024.113783>
- 16 Burgess, H. M. & Mohr, I. Cellular 5'-3' mRNA exonuclease Xrn1 controls double-stranded RNA accumulation and anti-viral responses. *Cell Host Microbe* **17**, 332-344 (2015). <https://doi.org/10.1016/j.chom.2015.02.003>
- 17 Dorrity, T. J. *et al.* Long 3'UTRs predispose neurons to inflammation by promoting immunostimulatory double-stranded RNA formation. *Sci Immunol* **8**, eadg2979 (2023). <https://doi.org/10.1126/sciimmunol.adg2979>
- 18 Koh, H. R., Kidwell, M. A., Raganathan, K., Doudna, J. A. & Myong, S. ATP-independent diffusion of double-stranded RNA binding proteins. *Proc Natl Acad Sci U S A* **110**, 151-156 (2013). <https://doi.org/10.1073/pnas.1212917110>
- 19 Koh, H. R., Kidwell, M. A., Doudna, J. & Myong, S. RNA Scanning of a Molecular Machine with a Built-in Ruler. *J Am Chem Soc* **139**, 262-268 (2017). <https://doi.org/10.1021/jacs.6b10387>

We thank all reviewers for the thoughtful comments both during the 1st and 2nd rounds of review. The 2nd round feedback focused entirely on the single-molecule analyses added during our 1st revision. To address these new comments, we performed additional single-molecule experiments and analyses, resulting in **14 new figure panels** (Fig. 3G, 3H, S5A-S5E, S6A-S6E, S8A-S8B) and **2 revised figure panels** (Fig. 3F, 4D).

Revised text was indicated red in Main text and figure legends.

In total (from both 1st and 2nd rounds of review), we have **added 30 new figure panels** and **revised 7 figure panels**.

We would like to emphasize that our conclusions are based on a combination of experimental approaches, including cellular, biochemical and structural data (Fig 1-6, Fig S1-S4, S7, S9-S10) as well as single molecule analyses (Fig 3, 4, S5, S6, S8). These multiple lines of evidence collectively support our original conclusions, which are now further strengthened by the additional data in this revised manuscript.

Reviewer #2 (Remarks to the Author)

Could the authors clarify the following sentence "Instead, they are consistent with rapid scanning motion along the dsRNA. (lines 289~290)." Could the authors explain in more detail why the fluctuations observed in their FRET experiment suggest there is rapid scanning? How would the fluctuations look like if a protein bound RNA but there was no scanning?

> We apologize for the lack of clarity in presenting our single molecule data. To address this, we have now provided a more detailed explanation of our results along with additional controls supporting PKR scanning motions. See main text page 7 (red fonts).

1. Anticorrelation between FRET donor and acceptor fluorescence:

The FRET fluctuations we observed here were accompanied by anticorrelated changes in FRET acceptor (PKR) and donor (dsRNA), which are indicative of their distance changes, not random noise. We have shown similar FRET changes with another dsRNA-binding protein TRBP and demonstrated that it also moves along dsRNA (PMID: 23251028, 27958730). In our earlier submission, the illustrations used for the fluorescence traces were too thick to clearly visualize this anticorrelation. We now provide thinner traces to clearly present the anticorrelated changes (Figure 3F, S5A, S5B).

2. Autocorrelation curve:

To further demonstrate that the FRET fluctuations are not random noise, we performed FRET autocorrelation analysis in the "off" (PKR/PACT unbound) phase and in the absence of PKR/PACT. Unlike in the "on" phase, where we observed a clear autocorrelation signal (exponential decay), no significant autocorrelation was detected in these control conditions, indicating that the observed fluctuations in the "on" state are not random noise. These new results are shown in Figure 3G, S5C, S5D, S6D.

3. Length dependence:

The decay rate of the autocorrelation function during the "on" phase was dependent on dsRNA length, with shorter dsRNA exhibiting faster decay (Figure 3G, 3H). This length dependence further supports the dsRNA scanning mechanism.

4. Comparison with other proteins:

Our observations are consistent with known 1D-diffusion behaviors of other dsRNA-binding proteins such as TRBP, PACT and Staufen 1 (PMID: 23251028, 27958730, 26184879). However, not all dsRNA-binding proteins exhibit this behavior. For instance, we found that Dicer and ADAR1/2 show little evidence of 1D diffusion on dsRNA (PMID:23251028, 26184879). These proteins serve as useful reference points for comparison with PKR, further supporting our interpretation that the FRET fluctuations observed in our study reflect genuine PKR scanning along dsRNA rather than nonspecific fluorescence noise.

Reviewer #3 (Remarks to the Author)

The authors have submitted a revised manuscript that has addressed some of this reviewer's concerns and introduce an intriguing diffusion model for the interactions between PKR and PACT. However, the quality of single-molecule analysis is insufficient, and the lack of appropriate controls raises concerns about the accuracy of the data presented. Specifically:

1) In the smFRET experiment, the anti-correlated nature of the donor and acceptor signals typically suggests energy transfer. However, in Figure 3F, this anti-correlation is too weak, particularly in the representative trajectory of PACT, where a decrease in the Cy3 signal does not correspond to an increase in the Cy5 signal. This does not appear to indicate a true FRET event.

>We apologize for the lack of clarity in presenting the FRET signal fluctuation. The FRET and fluorescence traces were too thick to readily visualize opposing changes in donor and acceptor fluorescence. We now provide thinner traces from many more binding events (in Figure 3F, 4D, S5A-S5E, S6B-S6C, S8A-S8B), which clearly show anticorrelated changes of FRET donor and acceptor signals. We also provide additional evidence and control experiments to support our model that PKR and PACT scan along dsRNA (see our response below).

2) Unlike Ref #53 and #54, the current manuscript does not demonstrate repetitive FRET fluctuations. The expected continuous and periodic movement of the Cy3-protein along the dsRNA should produce repetitive FRET fluctuations, rather than random variations. It is unclear how the authors distinguish between background FRET fluctuations and those caused by protein diffusion. The observed fluctuations might simply result from background noise. For instance, in Figure 3F, FRET efficiency varies significantly even when PACT is not bound. To strengthen their findings, the authors should include TRBP or other dsRNA-binding proteins to confirm that they are observing mobile binding rather than static interactions.

> This FRET fluctuation is not due to random variability of the fluorescent signal, as such signal shows no autocorrelation (Fig. 3G, S5D, S6D), whereas FRET fluctuations from the protein movement in fact show a clear autocorrelation Fig. 3G, S6D. Previous study from our lab (Figure 4E in PMID: 23251028) also showed movement of PACT (as well as TRBP), which is consistent with the current study. The seemingly noisier pattern observed in the current work could be likely due to the use of longer dsRNA (112 bp dsRNA), which is the optimal length to observe the impact of PACT on PKR in our kinase assay (Fig. 4A-4B). However, to more clearly demonstrate periodic movement of PKR along dsRNA, we now show traces with shorter dsRNAs (42 and 59 bp dsRNA) in Figure S5E.

3) Substrates with various length should be tested to confirm the diffusion of PKR and PACT, as demonstrated in Ref #53 and #54.

> Following reviewer's suggestions, we performed FRET experiments with 42, 59 and 112 bp dsRNA (these lengths were chosen to be relevant for PKR functions). Consistent with the notion that PKR scans along dsRNA, the decay rate of the autocorrelation function during the "on"

phase of PKR was dependent on dsRNA length, with shorter dsRNA exhibiting faster decay, further supporting a movement-based mechanism. These new results are shown in Figure 3G, 3H, S5E.

Reviewer #4 (Remarks to the Author)

In this work, the authors analyze how proteins involved in the innate immune response recognize and interact with RNA. This is a very complex paper and the focus of my review is going to be on the single molecule fluorescence aspects of the work.

The central issue is that the authors use single molecule assays (FRET) to provide evidence for a scanning mechanism (this is brought up multiple times in the abstract alone). Reviewer 3 shared concerns about this data and all reviewers shared concerns over how strongly the presented data support scanning. I agree with the reviewer assessments. As presented, the single molecule data is potentially consistent with scanning but does not exclude other models. As presented, it does not strongly support the authors conclusions and is not yet suitable for publication in Nature Communications.

The central flaw in the presentation of the single molecule data is that it is presented as scanning being the de facto mechanism. Instead, the single molecule data should be used to test a scanning model. As such many of the control experiments previously used by the Myong lab to justify scanning (and referenced by Rev 3) are missing in this manuscript. These include analysis using orthogonal methods (PIFE) and determining whether or not the sliding time is dependent on RNA length. These are particularly important since the FRET traces presented in the manuscript are much noisier than those seen in Ref. 54, for example.

>We thank the reviewer for these helpful suggestions. We now provide additional evidence, as summarized below, to further support movement of PKR and PACT on dsRNA. All experiments were performed and analyzed in a manner consistent with our earlier work from the Myong lab (Ref. 53, 54).

1. Anticorrelation between FRET donor and acceptor fluorescence:

The FRET fluctuations we observed here were accompanied by anticorrelated changes in FRET acceptor (PKR) and donor (dsRNA), which are indicative of their distance changes, not random noise. We have shown similar FRET changes with another dsRNA-binding protein TRBP and demonstrated that it also moves along dsRNA (PMID: 23251028, 27958730). In our earlier submission, the illustrations used for the fluorescence traces were too thick to clearly visualize this anticorrelation. We now provide thinner traces to clearly present the anticorrelated changes (Figure 3F, S5A, S5B).

2. Autocorrelation curve:

To further demonstrate that the FRET fluctuations are not random noise, we performed FRET autocorrelation analysis in the “off” (PKR/PACT unbound) phase and in the absence of PKR/PACT. Unlike in the “on” phase, where we observed a clear autocorrelation signal (exponential decay), no significant autocorrelation was detected in these control conditions, indicating that the observed fluctuations in the “on” state are not random noise. These new results are shown in Figure 3G, S5C, S5D, S6D.

3. Length dependence:

The decay rate of the autocorrelation function during the “on” phase was dependent on dsRNA length, with shorter dsRNA exhibiting faster decay (Figure 3G, 3H). This length dependence further supports the dsRNA scanning mechanism.

4. Comparison with other proteins:

Our observations are consistent with known 1D-diffusion behaviors of other dsRNA-binding proteins such as TRBP, PACT and Staufen 1 (PMID: 23251028, 27958730, 26184879). However, not all dsRNA-binding proteins exhibit this behavior. For instance, we found that Dicer and ADAR1/2 show little evidence of 1D diffusion on dsRNA (PMID:23251028, 26184879). These proteins serve as useful reference points for comparison with PKR, further supporting our interpretation that the FRET fluctuations observed in our study reflect genuine PKR scanning along dsRNA rather than nonspecific fluorescence noise.

Altogether these additional data further strengthen our conclusion that PKR and PACT scan along dsRNA.

In addition, there are several aspects of the single molecule data and analysis that should be revisited. First, the raw microscopy data, and ideally the fluorescence trajectories, need to be uploaded to a central repository so that the analysis could be reproduced by others. Second, virtually no information is included in the manuscript on how FRET peaks were identified. This is particularly relevant for Figure 3F. In the top trace, there is a spike in fluorescence at about 8s, why is this not an on event? In the bottom trace, there are numerous oscillations in FRET starting at about 8s and lasting until the described on event. Why are these not peaks?

> We will provide the raw microscopy data and fluorescence trajectories in a central repository to ensure accessibility and reproducibility, in accordance with *Nature Communications'* publication policies.

Regarding the selection of *on-state* events in the FRET traces, we now provide more detailed description in the main text (page 7) and Figure S5A legend. Briefly, we identified regions with high FRET ($> \sim 0.6$) and anticorrelated changes in FRET donor and acceptor signals. While some binding events were isolated (e.g. the event marked with * in Figure 3F), many occurred in clusters (“On”), during which rapid FRET fluctuations were observed, similar to those previously reported for TRBP and PACT^{53,54} (Figures 3F, S5B). We classified short-lived, isolated events (≤ 2 s) as abortive binding (common practice in the field, e.g. PMID: 27934859, 23251028) and excluded them from the autocorrelation analysis. The clustered events were analyzed.

In addition, the correlation analysis is potentially problematic given the noisiness of the data in Figure 3F. The correlation times the authors report are quite short (0.2 s) and this seems to be on a similar time scale as the oscillation of the background noise in Figure 3F with PACT. What would result of a correlation analysis of the FRET signals/noise of the “off times” when PACT or other proteins are not bound? In addition, to reiterate the point above, information is needed in the manuscript about how peaks were identified since evidently the correlation analysis only

included "200 binding events".

>We now show autocorrelation analysis for “off-states” for PKR and PACT, as well as in the absence of these proteins (Figure 3G, S5D, S6D). In all cases, autocorrelation analysis showed no decay curves, clearly distinct from the “on time” analysis results. Additionally, the sliding time of PACT (inverse of the decay rate in autocorrelation curve) is similar to that reported in Ref 54 (PMID: 23251028). Finally, we have explicitly added the exact number of binding events analyzed in each figure legend.

Finally, there are some book keeping items. First, the actual number of events analyzed in the correlation traces needs to be included (just saying over 200 is insufficient). Second, the authors should include the number of events observed and the number of AOIs/RNAs immobilized in each experiment. This is concerning a bit since I suspect the author's FOV is large with several hundred molecules per FOV. If only 200 events or so are analyzed, this suggests that binding may be somewhat rare. Third, I am highly suspect of single molecule experiments when only a handful of traces are shown. The authors should randomly select 10-20 traces from each analyzed experimental set and include them in supplemental so that readers have a better sense of the data quality. This is particularly relevant for the data in 3F with PACT which looks very noisy. It is a bit worrisome that the example trace may be the best "looking" trace they have.

> We have now explicitly stated the exact number of binding events analyzed in the figure legend. Regarding the relatively low frequency of PKR and PACT binding events, this reflects a deliberate choice to maintain physiological relevance. As shown in Figure 2 and 4, PACT can inhibit PKR without competing for dsRNA binding, both in vitro and in vivo. High concentrations of PKR or PACT would lead to artificial competition between these proteins for dsRNA, which does not accurately represent physiological conditions. Thus, we carefully selected the lowest protein concentrations that still allowed robust measurement of binding events. Based on these considerations, we performed our analysis at 50 nM PKR and PACT. Using these observations, we reliably showed the strong impact of PACT on PKR movement on dsRNA. To further address the reviewer's request, we have included additional representative traces for each condition in Figure S5, S6 and S8.

We thank for the comments. But

Reviewer #3 (Remarks to the Author):

I'd like to begin by emphasizing that a large volume of data does not necessarily make a conclusion more convincing. In fact, some of the results in the previous version, such as Figure 3F, raised more questions about the reliability of the experiments and analyses (also brought up by reviewer #4 multiple times and has since been removed). In the current manuscript, the anti-correlation between the donor and acceptor is more visible. The representative smFRET time course traces support the authors' conclusion that interactions between PKR and the 112 bp dsRNA are dynamic. The additional analysis and PIFE results further strengthen these findings.

However, some of the newly included data do not seem to align with the idea of PKR sliding. In Figure S5E, the FRET efficiency remained high and stable during the "On" state for PKR binding to the 59 bp and 42 bp dsRNA, similar to the behavior of Dicer and ADAR1/2 statically binding to dsRNA (as referenced by the authors, PMID: 23251028 and 26184879). To my understanding, given the contour lengths of the 59 bp and 42 bp dsRNAs (15 nm and 11 nm, respectively), a protein footprint of 15 bp (2 nm) diffusing along the entire length of the substrate should produce a large amplitude of FRET change. In fact, when TRBP diffused on the 38 bp and 19 bp dsRNA, rapid and more obvious FRET fluctuations were observed (again, as referenced by the authors, PMID: 23251028).

> We thank the reviewer for the comments. While we acknowledge that there are some differences in FRET fluctuation patterns for PKR vs. those previously reported for TRBP, both systems exhibit dsRNA length-dependent autocorrelation curves and PIFE signals. Combined with our biochemical assays—which demonstrate length-dependent activation without evident multimerization—we believe the simplest model to explain our single molecule data is a sliding model. The differences in fluctuation patterns may be attributable to factors such as the precise position of Cy5 label on the protein (which may affect the closest Cy3-Cy5 distance), as well as intrinsic differences in processivity in diffusion behavior (along the line of the reviewer's comment below).

If the authors insist that the time course traces in Figure S5E represent FRET fluctuations (which, to me, is not immediately apparent), they should discuss why the patterns differ so significantly between the two studies. For example, PKR cannot diffuse along the entire length of the dsRNA? If the authors do not interpret these as FRET fluctuations, they should also discuss why PKR seems to diffuse on the 112 bp dsRNA but not on the 59 bp or 42 bp dsRNAs.

> see above.

These observations on 59 bp or 42 bp dsRNAs raise additional concerns about the experimental design and the interpretations of the single-molecule data. For instance, multiple PKR molecules could potentially bind to the 112 bp substrates, but the authors might only be detecting the one closest to the RNA end (with PKR further away from the end falling outside the FRET

measurement range). This is plausible because the PKR concentration used in the single-molecule experiment was 50 nM, while 100 nM PKR efficiently formed dimers on 112 bp dsRNA (Figures 3 and 4). Given that PKR was labeled in the C-terminal kinase domain, which is linked to the RNA-binding domains by a large flexible linker, any conformational changes in the kinase domain—such as dimerization or dimer-dimer interactions—could potentially affect the FRET efficiency. I recognize that these concerns may not be easily addressed without further studies. But as noted by Reviewer #4, these data do not rule out other possibility and should at least be discussed in the manuscript. After all, strong claims require strong evidence.

> In our single molecule analysis, PKR binding typically occurs once or twice within a 30-second window at RT, making simultaneous occupancy by multiple molecules a rare event under these conditions. In contrast, the PKR activity assay in Figure 3 and 4 was done over the course of 10 min at 37°C. In addition, the fraction of activated PKR is not directly determined in these assays. Moreover, EMSA results (Figure S4A) indicate that on 42 and 62 bp dsRNA, significant accumulation of two PKR copies is observed only at much higher concentrations (625–1250 nM).